# Improving Graph Neural Networks by Learning Continuous Edge Directions

**Seong Ho Pahng**[1,2] **& Sahand Hormoz**[3,2,4*]
[1]Department of Chemistry and Chemical Biology, Harvard University
[2]Department of Data Science, Dana-Farber Cancer Institute
[3]Department of Systems Biology, Harvard Medical School
[4]Broad Institute of MIT and Harvard
`spahng@g.harvard.edu, sahand_hormoz@hms.harvard.edu`

## Abstract

Graph Neural Networks (GNNs) traditionally employ a message-passing mechanism that resembles diffusion over undirected graphs, which often leads to homogenization of node features and reduced discriminative power in tasks such as node classification. Our key insight for addressing this limitation is to assign fuzzy edge directions—that can vary continuously from node $i$ pointing to node $j$ to vice versa—to the edges of a graph so that features can preferentially flow in one direction between nodes to enable long-range information transmission across the graph. We also introduce a novel complex-valued Laplacian for directed graphs with fuzzy edges where the real and imaginary parts represent information flow in opposite directions. Using this Laplacian, we propose a general framework, called Continuous Edge Direction (CoED) GNN, for learning on graphs with fuzzy edges and prove its expressivity limits using a generalization of the Weisfeiler-Leman (WL) graph isomorphism test for directed graphs with fuzzy edges. Our architecture aggregates neighbor features scaled by the learned edge directions and processes the aggregated messages from in-neighbors and out-neighbors separately alongside the self-features of the nodes. Since continuous edge directions are differentiable, they can be learned jointly with the GNN weights via gradient-based optimization. CoED GNN is particularly well-suited for graph ensemble data where the graph structure remains fixed but multiple realizations of node features are available, such as in gene regulatory networks, web connectivity graphs, and power grids. We demonstrate through extensive experiments on both synthetic and real graph ensemble datasets that learning continuous edge directions significantly improves performance both for undirected and directed graphs compared with existing methods. Our code is available on GitHub.

## 1 Introduction

Graph Neural Networks (GNNs) have emerged as a powerful tool for learning from data that is structured as graphs, with applications ranging from social network analysis to molecular chemistry (Kipf & Welling, 2017; Zhou et al., 2020; Gilmer et al., 2017). GNNs typically employ a message passing mechanism where nodes aggregate and then transform feature information from their neighbors at each layer, enabling them to learn node representations that capture both local and global graph structures. When the graph is undirected, the aggregation of node features mimics a diffusion process. Each node's representation becomes the averaged features of its immediate neighbors, leading to a homogenization of information across the graph. As depth increases, this diffusion of information culminates in a uniform state where node representations converge towards a constant value across all nodes, which severely limits the discriminative power of GNNs, especially in tasks such as node classification (Rusch et al., 2023a; Oono & Suzuki, 2020; Cai & Wang, 2020; Li et al., 2018a; Keriven, 2022; Chen et al., 2020a; Wu et al., 2023; 2024).

---

*Corresponding author.

Our key insight for improving the performance of GNNs is to alter the nature of information transmission between nodes from diffusion to flow. To do so, we add directions to the edges of a graph so that features can be propagated from node $v_i$ to its neighbor node $v_j$ without reciprocal propagation of information from node $v_j$ to node $v_i$. Unlike diffusion, where information uniformly spreads across available paths, flow is directional and preserves the propagation of information across longer distances within a graph, as illustrated in Figure 1. In general, the optimal information propagation could require edges whose directions fall anywhere in the continuum of node $v_i$ pointing to node $v_j$ to vice versa.

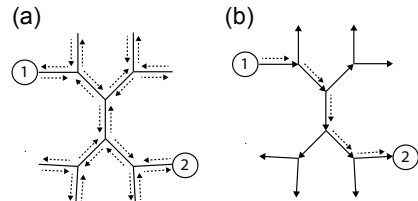

Figure 1: (a) When edges are undirected, information diffuses across the graph and long-range transmission of information between nodes 1 and 2 is not possible. (b) Once the optimal edge directions are learned, information can flow directly from node 1 to node 2.

To capture such continuous edge directions, we propose a concept of 'fuzzy edges,' where the direction of an edge between any two nodes $v_i$ and $v_j$ is not a discrete but a continuous value. An edge's orientation can range continuously—from exclusively pointing from node $v_i$ to node $v_j$, through a fully bidirectional state, to exclusively pointing from node $v_j$ to node $v_i$. Therefore, 'fuzzy' direction essentially controls the relative amount of information flow from node $v_i$ to node $v_j$ and the reciprocal flow from node $v_j$ to node $v_i$. To effectively model this directional flexibility, we introduce a complex-valued graph Laplacian called a fuzzy Laplacian. In this framework, the real part of the $ij$-th entry in the fuzzy Laplacian matrix quantifies the degree of information transmission from node $v_j$ to node $v_i$, while the imaginary part measures the flow from node $v_i$ to node $v_j$.

Next, we introduce the Continuous Edge Direction (CoED) GNN architecture. At each layer, a node's neighbors' features are scaled by the directions of their connecting edges and aggregated. This aggregation is performed separately for incoming and outgoing edges, following Rossi et al. (2024), resulting in distinct features for incoming and outgoing messages. This is implemented by applying the fuzzy Laplacian to the node features, where the real and imaginary parts correspond to the features aggregated from incoming neighbors and outgoing neighbors, respectively. These aggregated features are then affine transformed using learnable weights and combined with the node's own transformed features. A nonlinear activation function is applied to obtain the updated node features. This process is repeated for each layer. The continuous edge directions have the added benefit that they are differentiable. During training, both the edge directions and weight matrices are learned simultaneously using gradient-based optimization to improve the learning objective.

Importantly, our approach is fundamentally different from methods such as Graph Attentions Network (GAT) (Veličković et al., 2018) or graph transformers (Dwivedi & Bresson, 2021; Rampášek et al., 2022) that learn attention coefficients to assign weights to each edge of the graph based on the features (and potentially the positional encoding) of the nodes connected by that edge. While the attention mechanism can capture asymmetric relationships by computing direction-specific attention weights based on node features, they do not learn edge directions as independent parameters. Instead, the attention coefficients from node $v_i$ to node $v_j$ are functions of the features of $v_i$ and $v_j$, and will change if node features change. In contrast, our approach introduces continuous edge directions as learnable parameters that are optimized end-to-end, independent of the node features.

Learning edge directions can make CoED susceptible to overfitting, especially when the same graph is used for both training and testing, as in standard node classification tasks. In such cases, edge directions may be optimized for the training nodes at the expense of effective information flow to the test nodes that are withheld during training. In contrast, CoED is very effective on graph ensemble data, where the graph structure remains fixed but multiple realizations of node features and targets (such as node labels) are available. This allows optimization of information flow across all edges simultaneously without masking parts of the graph for training and testing. Instead, training and testing splits are based on different feature realizations rather than on subsets of the graph. Graph ensemble data are increasingly common across various domains. For example, in biology, gene-regulatory networks are constant directed graphs where nodes represent genes and edges represent gene-gene interactions, while node features like gene expression levels vary across different cells. Similarly, in web connectivity, the network of websites remains relatively static, but traffic patterns change over time, providing different node feature sets on the same underlying graph. In power

grids, the network of electrical components is fixed, while the steady-state operating points of these components vary under different conditions, yielding multiple observations on the same graph. In all these cases, a fixed graph is paired with numerous feature variations. By applying CoED GNN to these scenarios, we demonstrate that learning edge directions significantly improves performance for both directed and undirected graphs. The main contributions of this paper are the following:

- We introduce a principled complex-valued graph Laplacian for graphs where edge directions can vary continuously and prove that it is more expressive than existing forms of Laplacians for directed graphs, such as the magnetic Laplacian.
- We propose an architecture called Continuous Edge Direction (CoED) GNN, which is a general framework for learning on directed graphs with fuzzy edges. We prove that CoED GNN is as expressive as a weak form of the Weisfeiler-Leman (WL) graph isomorphism test for directed graphs with fuzzy edges.
- Using extensive experiments, we show that learning edge directions significantly improves performance by applying CoED GNN to both synthetic and real graph ensemble data.

## 2 PRELIMINARIES

A graph is defined as a pair $\mathcal{G} = (\mathcal{V}, \mathcal{E})$, where $\mathcal{V} = \{v_1, v_2, \ldots, v_N\}$ is a set of $N$ nodes, and $\mathcal{E} \subseteq \mathcal{V} \times \mathcal{V}$ is a set of edges connecting pairs of nodes. Each node $v_i$ is associated with a feature vector $\mathbf{f}_i \in \mathbb{R}^D$, where $D$ is the dimensionality of the feature space, and collectively these feature vectors form the node feature matrix $\mathbf{F} \in \mathbb{R}^{N \times D}$. Additionally, each node is assigned a prediction target, such as a class label for classification tasks or a continuous value for regression tasks.

The connectivity of a graph is encoded in an adjacency matrix $\mathbf{A} \in \{0, 1\}^{N \times N}$. If there is an undirected edge between $v_i$ and $v_j$, then both $A_{ij} = 1$ and $A_{ji} = 1$. For a directed edge, one of $A_{ij}$ or $A_{ji}$ is 1 while the other is 0, specifying the direction of information flow. A directed edge, $A_{ij} = 1$ and $A_{ji} = 0$, indicates that $v_j$ sends information to $v_i$. Hence, we refer to $v_j$ as the in-neighbor of $v_i$ and conversely to $v_i$ as the out-neighbor of $v_j$. If both $A_{ij} = 0$ and $A_{ji} = 0$, there is no edge between $v_i$ and $v_j$. Accordingly, in a directed graph, we define two distinct degree matrices: the in-degree matrix $\mathbf{D_{in}} = \mathrm{diag}(\mathbf{A1})$ and the out-degree matrix $\mathbf{D_{out}} = \mathrm{diag}(\mathbf{A}^\top \mathbf{1})$.

GNNs iteratively processes node features $\mathbf{F}$ via message-passing mechanism that leverages the structural information of the graph $\mathcal{G}$. This process involves two main steps at each layer $l$:

1. Message Aggregation: For each node $v_i$, an aggregated message $\mathbf{m}_{i,\mathcal{N}(i)}^{(l)}$ is computed from the features of its neighbors:

$$\mathbf{m}_{i,\mathcal{N}(i)}^{(l)} = \mathrm{AGGREGATE}\left(\{\!\!\{(\mathbf{f}_i^{(l-1)}, \mathbf{f}_j^{(l-1)}) \mid j \in \mathcal{N}(i)\}\!\!\}\right)$$

Here, $\mathcal{N}(i)$ denotes the set of nodes $v_j$ that are connected to node $v_i$ by an edge.

2. Feature Update: The feature vector of node $v_i$ is then updated using the aggregated message:

$$\mathbf{f}_i^{(l)} = \mathrm{UPDATE}\left(\mathbf{f}_i^{(l-1)}, \mathbf{m}_{i,\mathcal{N}(i)}^{(l)}\right)$$

AGGREGATE and UPDATE are functions with learnable parameters, and their specific implementations define different GNN architectures (Gilmer et al., 2017).

## 3 FORMULATION OF GNN ON DIRECTED GRAPHS WITH FUZZY EDGES

### 3.1 CONTINUOUS EDGE DIRECTIONS AS PHASE ANGLES

To describe a continuously varying edge direction between node $v_i$ and node $v_j$, we assign an angle $\theta_{ij} \in [0, \pi/2]$ to the edge connecting $v_i$ to $v_j$. During aggregation of features from neighbors, features propagated from $v_j$ to $v_i$ are scaled by a factor of $\cos \theta_{ij}$. Conversely, the features that $v_j$ receives from $v_i$ are scaled by $\sin \theta_{ij}$. For example, when $\theta_{ij} = 0$, we have a directed edge where $v_j$ sends messages to $v_i$ but does not receive any messages from $v_i$. When $\theta_{ij} = \pi/4$, the edge is undirected and the same scaling is applied to the messages sent and received by $v_i$ to and from $v_j$,

i.e., $\cos \pi/4 = \sin \pi/4 = 1/\sqrt{2}$. To ensure consistency, we require that the message received by $v_i$ from $v_j$ should be equivalent to the message sent by $v_j$ to $v_i$. It follows that $\theta_{ji} = \pi/2 - \theta_{ij}$. We define the phase matrix $\Theta \in [0, \pi/2]^{N \times N}$ to describe the directions of all the edges in a graph. $(\Theta)_{ij}$ is only defined if there is an edge connecting nodes $v_i$ and $v_j$.

## 3.2 Fuzzy Graph Laplacian

To keep our message-passing GNN as expressive as possible, we define a Laplacian matrix that, during the aggregation step, propagates information along directed edges but keeps the aggregated features from in-neighbors and out-neighbors for each node distinct by assigning them to the real and imaginary parts of a complex number, respectively. For a given $\Theta$, we construct the corresponding fuzzy graph Laplacian $\mathbf{L}_F$ as follows. The diagonal entries $(\mathbf{L}_F)_{ii}$ are zero as we cannot define edge directions for self-loops, and off-diagonal entries are either zero or a phase value,

$$(\mathbf{L}_F)_{ij} = \begin{cases} 0 & \text{if } A_{ij} = A_{ji} = 0 \\ \exp(\mathrm{i}\theta_{ij}) & \text{otherwise} \end{cases} \tag{1}$$

Since $\theta_{ij}$ and $\theta_{ji}$ are related by $\theta_{ji} = \pi/2 - \theta_{ij}$, it follows that $\mathbf{L}_F = \mathrm{i}\mathbf{L}_F^\dagger$, where $\dagger$ is the conjugate transpose. $\mathrm{Re}[\mathbf{L}_F]$ thus encodes all $i \leftarrow j$ edges scaled by $\cos \theta_{ij}$ and $\mathrm{Im}[\mathbf{L}_F]$ all $i \rightarrow j$ edges scaled by $\sin \theta_{ij}$. In Appendix D, we show the fuzzy graph Laplacian admits orthogonal eigenvectors with eigenvalues of the form $a + \mathrm{i}a$ with $a \in \mathbb{R}$. Therefore, the eigenvectors of our Laplacian provide positional encodings that are informed by the directions of the edges in addition to their connectivities. We provide the visualizations of the eigenvectors in Appendix C. A key implication of our Laplacian for GNN architectures is stated in the following theorem.

**Theorem 1.** *A message-passing GNN whose aggregation step is performed using the fuzzy graph Laplacian is as expressive as the weak form of the Weisfeiler-Leman (WL) graph isomorphism test for directed graphs with fuzzy edges.*

We prove this theorem in Appendix E. The most commonly used Laplacian for directed graphs is the magnetic Laplacian (Shubin, 1994; Furutani et al., 2020; Zhang et al., 2021; He et al., 2022a), which is also complex-valued. For directed graphs with fuzzy edges, the magnetic Laplacian is not as expressive as the Laplacian proposed above. We provide a proof in Appendix F. Briefly, the real and imaginary parts of the aggregated features produced by the magnetic Laplacian are both linear combinations of in- and out- neighbor messages. In principle, GNNs should be able to disentangle these linear combinations to recover the in- and out- neighbor messages. However, the linear combinations depend on the local neighborhood of each node which is distinct from one node to another, whereas GNN parameters are shared across all nodes. Therefore, in general, a GNN using the magnetic Laplacian loses the ability to disentangle the in- and out- neighbor messages at each node and thus has lower expressivity. Our Laplacian does not suffer from this limitation since by construction the real and imaginary values uniquely correspond to the in- and out- neighbor aggregated messages, respectively.

## 3.3 Model Architecture: Continuous Edge Direction (CoED) GNN

To ensure maximum expressivity, a message-passing mechanism on a directed graph should, for each node, separately aggregate the features of the in-neighbors and the out-neighbors, and independently process the two types of aggregated features and the self-features to obtain an updated feature for each node. To this end, we define in- and out- edge weight matrices as $\mathbf{A}_\leftarrow = \mathrm{Re}[\mathbf{L}_F]$ and $\mathbf{A}_\rightarrow = \mathrm{Im}[\mathbf{L}_F]$, respectively. We compute the in- and out- degree matrices as $\mathbf{D}_\leftarrow = \mathrm{diag}\,(\mathbf{A}_\leftarrow \mathbf{1})$ and $\mathbf{D}_\rightarrow = \mathrm{diag}\,(\mathbf{A}_\rightarrow \mathbf{1})$, respectively. Following Rossi et al. (2024), but extending it to graphs with continuous edge directions, we define in- and out- fuzzy propagation matrices as

$$\mathbf{P}_\leftarrow = \mathbf{D}_\leftarrow^{-1/2} \mathbf{A}_\leftarrow \mathbf{D}_\rightarrow^{-1/2} \quad \text{and} \quad \mathbf{P}_\rightarrow = \mathbf{D}_\rightarrow^{-1/2} \mathbf{A}_\rightarrow \mathbf{D}_\leftarrow^{-1/2} \tag{2}$$

Using these matrices, we compute in- and out- messages at layer $l$ as

$$\mathbf{m}_\leftarrow^{(l)} = \mathbf{P}_\leftarrow \mathbf{F}^{(l-1)} \quad \text{and} \quad \mathbf{m}_\rightarrow^{(l)} = \mathbf{P}_\rightarrow \mathbf{F}^{(l-1)} \tag{3}$$

which defines the AGGREGATE function.

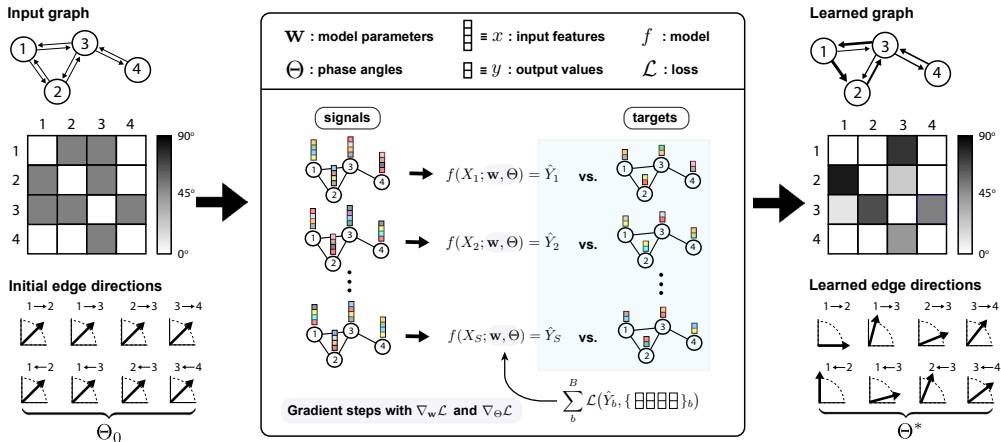

Figure 2: Schematic of training with a graph ensemble data. The input graph is undirected (left box). The graph ensemble data contains multiple realizations of node features and corresponding target values, either at the node, edge, or graph level. The phase angle formulation allows continuous edge directions to be optimized alongside the GNN parameters in an end-to-end manner (middle box). The learned edge directions (right box) enable long range information transmission across the graph.

Since self-loops are omitted from $\mathbf{L}_F$, we include current node features along with the two directional messages in UPDATE function and update node features as,

$$\mathbf{F}^{(l)} = \sigma\big(\mathbf{F}^{(l-1)}\mathbf{W}^{(l)}_{\text{self}} + \mathbf{m}^{(l)}_{\leftarrow}\mathbf{W}^{(l)}_{\leftarrow} + \mathbf{m}^{(l)}_{\rightarrow}\mathbf{W}^{(l)}_{\rightarrow} + \mathbf{B}^{(l)}\big) \tag{4}$$

where $\sigma$ is an activation function, and $\mathbf{W}^{(l)}_{\text{self}/\leftarrow/\rightarrow}$ and $\mathbf{B}^{(l)}$ are self/in/out weight matrices and a bias matrix, respectively.

The features at the final layer are then transformed using a linear layer to obtain the output for a specific learning task. We use end-to-end gradient-based optimization to iteratively update both the phase matrix $\Theta$ and the GNN parameters $\mathbf{W}^{(l)}_{\text{self}/\leftarrow/\rightarrow}$ and $\mathbf{B}^{(l)}$ at each layer, as illustrated in Figure 2. We allow for the option to learn a different set of edge directions at each layer, $\Theta^{(l)}$, just as we have distinct GNN parameters at each layer. We provide a runtime analysis of CoED against other GNNs in Appendix G.

## 4 RELATED WORK

The issue of feature homogenization in GNNs, known as the oversmoothing problem, has been a significant concern. Early studies identified the low-pass filtering effect of GNNs (Defferrard et al., 2016; Wu et al., 2019), linking it to oversmoothing and loss of discriminative power (Li et al., 2018a; Oono & Suzuki, 2020). Proposed solutions include regularization techniques like edge dropout (Rong et al., 2020), feature masking (Hasanzadeh et al., 2020), layer normalization (Zhao & Akoglu, 2020), incorporating signed edges (Derr et al., 2018), adding residual connections (Chen et al., 2020b), gradient gating (Rusch et al., 2023b), and constraining the Dirichlet energy (Zhou et al., 2021). Dynamical systems approaches have also been explored, modifying message passing via nonlinear heat equations (Eliasof et al., 2021), coupled oscillators (Rusch et al., 2022), and interacting particle systems (Wang et al., 2022; Di Giovanni et al., 2023). Other methods involve learning additional geometric structures, such as cellular sheaves (Bodnar et al., 2022).

Extending GNNs to directed graphs has been addressed through various methods. GatedGNN (Li et al., 2016) processed messages from out-neighbors in directed graphs. Some works constructed symmetric matrices from directed adjacency matrices and their transposes to build standard Laplacians (Tong et al., 2020b; Kipf & Welling, 2017), while others (Ma et al., 2019; Tong et al., 2020a) developed Laplacians based on random walks and PageRank (Duhan et al., 2009). MagNet (Zhang et al., 2021) utilized the magnetic Laplacian to represent directed messages—a technique that has also been used to adapt transformers to directed graphs (Geisler et al., 2023) and to visualize them

(Fanuel et al., 2018). FLODE (Maskey et al., 2023) employed asymmetrically normalized adjacency matrices within a neural ODE framework. DirGNN (Rossi et al., 2024) separately processed the messages from in-neighbors and out-neighbors using asymmetrically normalized adjacency matrices, improving node classification on heterophilic graphs. A similar strategy was used in Koke & Cremers (2024), replacing the adjacency matrices with filters representing Faber polynomials. Recent graph PDE-based models (Eliasof et al., 2024; Zhao et al., 2023) introduced an advection term to model directional feature propagation alongside diffusion, assigning edge weights based on computed velocities between nodes, akin to attention coefficients in GAT. Finally, our approach is conceptually related to He et al. (2022b) where imbalance of incoming and outgoing messages across subsets of nodes is used to learn node embeddings for clustering.

GNNs have been increasingly applied in domains relevant to our work. In single-cell biology, GNNs have been used to predict perturbation responses in gene expression data (Roohani et al., 2023; Molho et al., 2024), with datasets compiled in scPerturb (Peidli et al., 2024). In web traffic analysis, a form of spatiotemporal data on graphs, GNNs often model temporal signals using recurrent neural networks on graphs (Li et al., 2018b; Chen et al., 2018; Sahili & Awad, 2023), with datasets and benchmarks provided by PyTorch Geometric Temporal library (Rozemberczki et al., 2021). In power grids, GNNs have been applied to predict voltage values (Ringsquandl et al., 2021) and solve optimal power flow problems (Donon et al., 2020; Böttcher et al., 2023; Piloto et al., 2024), with datasets compiled by Lovett et al. (2024).

## 5 EXPERIMENTS

### 5.1 NODE CLASSIFICATION WITHOUT EDGE DIRECTION LEARNING

While node classification is not the primary focus of our paper, we benchmarked our method on eleven standard datasets—including both undirected and directed graphs covering a wide range of sizes and homophily levels—to highlight the advantage of our Laplacian over alternative forms of Laplacians for directed graphs as well as the benefit of processing self, in-neighbor, and out-neighbor features separately. Importantly, we do not learn edge directions in this case, and hence the phase value is either $0$ or $\pi/2$ for directed graphs and $\pi/4$ for undirected graphs. For comparison, we include the classical models: GCN (Kipf & Welling, 2017), SAGE (Hamilton et al., 2017), GAT (Veličković et al., 2018); heterophily-specific model, GGCN (Yan et al., 2021); directionality-aware models: MagNet (Zhang et al., 2021), FLODE (Maskey et al., 2023), DirGNN (Rossi et al., 2024); and a model that learns geometric structure of graph, Sheaf (Bodnar et al., 2022). We also include a model based on a Laplacian for directed graphs constructed from the transition matrix of the graph by Chung (2005), and Cooperative GNNs (Finkelshtein et al., 2023), which classify a node as broadcasting, listening, both, or neither based on its own and its neighbors' features. Finally, we also include MLP to highlight the effect of solely processing the nodes' self-features without aggregating features across the graph. Further details of the datasets and hyperparameters are provided in Appendix A.1

As shown in Table 1, CoED demonstrates competitive performance across all eleven datasets, ranking within the top three in terms of test accuracy for most. While all models exhibit comparable results on Cora and Citeseer—which are undirected and homophilic—their performances differ significantly on the directed, heterophilic graphs. The classical models developed for undirected graphs particularly struggle on these datasets, with the exception of SAGE. This is because processing only the node's own features yields good performance, as evidenced by the MLP's results. In contrast, for the Squirrel and Chameleon datasets, processing directed messages along only one direction is crucial for good performance. Only FLODE, DirGNN, and CoED exhibit strong results on these datasets when configured accordingly. Specifically, for CoED, we introduce the $\alpha$ hyperparameter as in Rossi et al. (2024) to weigh the directional messages post aggregation, replacing $\mathbf{m}_{\leftarrow}^{(l)}\mathbf{W}_{\leftarrow}^{(l)} + \mathbf{m}_{\rightarrow}^{(l)}\mathbf{W}_{\rightarrow}^{(l)}$ in Equation 4 with $\alpha\mathbf{m}_{\leftarrow}^{(l)}\mathbf{W}_{\leftarrow}^{(l)} + (1-\alpha)\mathbf{m}_{\rightarrow}^{(l)}\mathbf{W}_{\rightarrow}^{(l)}$. In addition, we make the transformation of self-features optional.

Importantly, our results highlight the advantage of the fuzzy Laplacian over the magnetic Laplacian and the Chung Laplacian. In particular, the magnetic Laplacian does not process the aggregated messages from out-neighbors and in-neighbors separately. Instead, it combines them into both the real and imaginary components of the aggregated feature vector, thus losing the opportunity to pro-

| | Roman-Empire | SNAP-Patents | Texas | Wisconsin | Arxiv-Year | Squirrel | Chameleon | Citeseer | Computers | Photo | Cora |
|---|---|---|---|---|---|---|---|---|---|---|---|
| Hom. level | 0.05 | 0.07 | 0.11 | 0.21 | 0.22 | 0.22 | 0.23 | 0.74 | 0.78 | 0.81 | 0.81 |
| Undirected | ✗ | ✗ | ✗ | ✗ | ✗ | ✗ | ✗ | ✓ | ✓ | ✓ | ✓ |
| MLP | 64.94±0.62 | 31.34±0.05 | 80.81±4.75 | 85.29±3.31 | 36.70±0.21 | 37.53±1.74 | 39.05±3.74 | 74.02±1.90 | 83.56±0.26 | 90.75±0.31 | 75.69±2.00 |
| GCN | 73.69±0.74 | 51.02±0.06 | 55.14±5.16 | 51.76±3.06 | 46.02±0.26 | 39.47±1.47 | 40.89±4.12 | 76.50±1.36 | 89.65±0.52 | 92.70±0.20 | 86.98±1.27 |
| SAGE | 85.74±0.67 | 48.43±0.21 | 82.43±6.14 | 81.18±5.56 | 52.94±0.14 | 36.09±1.99 | 37.77±4.14 | 76.04±1.30 | 91.20±0.29 | 94.59±0.14 | 86.90±1.04 |
| GAT | 80.87±0.30 | 45.92±0.22 | 52.16±6.63 | 49.41±4.09 | 46.05±0.51 | 35.62±2.06 | 39.21±3.08 | 76.55±1.23 | 90.78±0.13 | 93.87±0.11 | 86.33±0.48 |
| GGCN | 74.46±0.54 | OOM | 84.86±4.55 | 86.86±3.29 | OOM | 37.46±1.57 | 38.71±3.04 | 77.14±1.45 | 91.81±0.20 | 94.50±0.11 | 87.95±1.05 |
| FLODE | 74.97±0.53 | OOM | 77.57±5.28 | 80.20±3.56 | OOM | 38.63±1.68 | 42.85±3.89 | 78.07±1.62 | 90.88±0.23 | 95.93±0.20 | 86.44±1.17 |
| Sheaf | 77.94±0.53 | OOM | 85.95±5.51 | 89.41±4.74 | 48.77±0.20 | 39.03±1.73 | 41.98±3.42 | 77.14±1.85 | 90.56±0.13 | 95.01±0.17 | 87.30±1.15 |
| MagNet | 88.07±0.27 | OOM | 83.3±6.1 | 85.7±3.2 | 60.29±0.27 | 42.7±1.5 | 44.5±1.1 | 75.26±1.63 | 90.30±0.27 | 94.54±0.19 | 82.63±1.80 |
| Chung | 87.35±0.53 | 64.77±0.23 | 80.54±4.65 | 81.79±5.42 | 53.01±0.45 | 42.46±1.77 | 43.47±3.64 | 76.08±1.11 | 92.57±0.16 | 95.47±0.14 | 86.03±1.63 |
| DirGNN | 91.23±0.32 | 73.95±0.05 | 83.78±2.70 | 85.88±2.11 | 64.08±0.26 | 44.19±2.42 | 46.08±2.67 | 76.63±1.51 | 92.97±0.26 | 96.13±0.12 | 86.27±1.45 |
| Co-GNN | 91.57±0.32 | 48.31±0.15 | 83.51±5.19 | 86.47±3.77 | 49.82±0.24 | 39.85±1.15 | 41.92±4.03 | 76.49±1.40 | 92.76±0.22 | 95.95±0.14 | 87.44±0.85 |
| CoED | 92.17±0.29 | 74.67±0.02 | 84.59±4.53 | 87.84±3.70 | 64.59±0.20 | 45.50±1.62 | 47.27±3.62 | 77.14±1.57 | 92.88±0.15 | 95.83±0.12 | 87.02±1.01 |

Table 1: Comparison of baseline models and CoED (without edge direction learning) for node classification task across different types of graphs. Top three models are colored by First, Second, Third. The reported numbers are the mean and standard deviation of test accuracies across different splits. The first two rows report the homophily ratios of the graphs and whether they are directed or undirected. OOM indicates out-of-memory error.

cess the two separately. Moreover, during the Laplacian convolution, the directed messages further mix with self-features encoded in the real component. This results in poor performance by MagNet on directed graphs. The real-world directed graphs are often not strongly connected and thus the corresponding transition matrices' top left singular vectors are not as informative as in the undirected cases. Since the Chung Laplacian is constructed from this singular vector, it shows relatively poor performance on such datasets as Roman-Empire or Wisconsin compared to the other directed models and instead delivers better results on datasets with undirected graphs. Sheaf also suffers on the datasets with directed graphs despite expanding the feature dimensions via an object called a stalk, because the sheaf Laplacian is constrained to be symmetric, thereby losing the ability to process directed messages. Taken together, our benchmarking demonstrates that CoED's ability to effectively process self-features and separately aggregate in-neighbor and out-neighbor messages using our fuzzy Laplacian enables it to achieve competitive performance across diverse datasets.

## 5.2 NODE REGRESSION ON GRAPH ENSEMBLE DATASET

Our key contribution is the joint learning of continuous edge directions alongside GNN parameters. This approach is particularly effective on graph ensemble data, where the graph structure remains fixed but multiple realizations of node features and targets exist. By learning edge directions for all edges without a need to mask parts of the graph, our method optimizes information flow across the entire graph. Learning edge directions for the node classification task above, where a subset of nodes are masked for testing, would optimize the edges connected to the training node at the expense of those connected to the test nodes, diminishing overall performance. However, in the graph ensemble setting, learning continuous edge directions substantially improves performance, as we empirically demonstrate below on both synthetic and real-world datasets.

### 5.2.1 SYNTHETIC DATASETS

**Directed flow on triangular lattice.** We begin by applying CoED GNN to a node regression problem constructed on a graph with continuous edge directions, where the target node features are obtained by directionally message-passing the input node features over long distances across the graph. To generate such a graph with continuous edge directions exhibiting long-range order, we created a two-dimensional triangular lattice, assigning each node a position in the 2d plane. We then defined a potential energy function $V$ on this plane, consisting of one peak and one valley (Figure 3(a)). The gradient of $V$ yields a vector field with long-range order, which we used to assign continuous edge directions to the edges of the triangular lattice (Figure 3(b)). Using this graph, we performed the message passing step of Equation 4 iteratively 10 times—using random matrices $\mathbf{W}_\rightarrow$, $\mathbf{W}_\leftarrow$, and $\mathbf{W}_{\text{self}}$ that were shared across all 10 iterations, starting from the initial node features to obtain the target node values. We repeated this procedure 500 times for different random initial node features and generated an ensemble of input node features and corresponding target node values. During training, we provided all models with the undirected version of the triangular lattice graph (i.e., all $\theta_{ij} = \pi/4$ for CoED). The goal of the learning task is to predict the target node values from the input node features. Additionally, CoED GNN is expected to learn

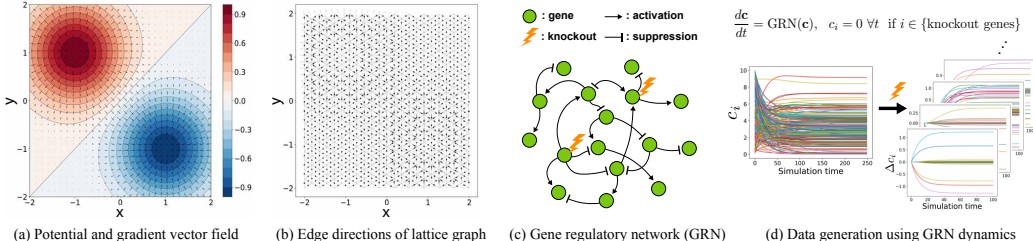

Figure 3: Synthetic datasets. (a-b) Triangular lattice graph with edge directions derived from the gradient of a 2d potential function $V$ (shown in a), creating long-range flows across the graph. (c-d) Gene regulatory network (GRN) represented as a directed graph where nodes are genes and edges denote interactions. Steady-state gene expression levels are obtained from GRN dynamics, with perturbations simulated by setting the expression levels of specific genes to zero.

the underlying ground truth continuous edge directions of the graph as part of its training. Further details on data generation are provided in Appendix A.2.1.

**Gene Regulatory Network (GRN) dynamics.** Gene Regulatory Networks (GRNs) are directed graphs where nodes represent genes and edges represent interactions between pairs of genes (Figure 3(c)). In these networks, when two genes interact, one either activates or suppresses the other. We used Hill functions with randomly chosen parameters to define the dynamics of these gene-gene interactions. We constructed a directed GRN graph with 200 nodes and randomly assigned interactions between them. Starting from random initial expression levels, we solved the system of nonlinear ordinary differential equations representing the GRN dynamics to obtain the steady-state expression levels of all genes. Next, we modeled gene perturbations by setting the expression levels of either one or two genes (the perturbed set of genes) to zero and recomputing the steady-state expression levels for all genes using the same GRN dynamics (Figure 3(d)). We performed this procedure for all single-gene perturbations and a subset of double-gene perturbations, resulting in 1,200 different realizations. Our learning task is to predict the steady-state expression levels of all genes following perturbation (target node values) given the initial steady-state expression levels with the perturbed genes set to zero (input node features). We provided baseline models with the original graph and CoED with the undirected version of the graph. Further details of the data generation are provided in Appendix A.2.2.

**Results.** Table 2 shows the test performances of CoED alongside several baseline models: classical models, GCN and GAT; a transformer-based model with positional encoding, GraphGPS (Rampášek et al., 2022); a directionality-aware model, MagNet and the model based on the Chung Laplacian; a higher-order model, DRew (Gutteridge et al., 2023); and a combination of directionality-aware and higher-order model, FLODE. Details of the training setup, hyperparameter search procedure, and selected hyperparameters are provided in Appendix A.2.2.

To identify which aspects of GNNs are particularly effective for learning on graph ensemble datasets, we analyze the baseline models' results in detail. On the undirected lattice graph, MagNet provides only a slight improvement over GCN, which is expected since MagNet reduces to ChebNet (Defferrard et al., 2016) on undirected graphs, and GCN is a first-order truncation of ChebNet. However, in the directed GRN experiments, MagNet shows substantial improvement over GCN. We also observe that higher-order GNNs like DRew and FLODE perform competitively on the undirected lattice graph. Notably, FLODE's instantaneous enhancement of connectivity via fractional powers of the graph Laplacian outperforms DRew's more gradual incorporation of higher-hop messages. However, both methods struggle on the directed GRN graph. The model based on the Chung Laplacian demonstrates strong performance on the lattice graph but offers only a modest improvement over GCN on the directed GRN. On both synthetic graphs, attention-based models—GAT and GraphGPS—deliver strong performance, coming in just behind CoED. GraphGPS, in particular, seems to benefit from its final global attention step, similar to how FLODE benefits from densifying the graph. We also notice that increasing the dimension of Laplacian positional encoding does not further enhance GraphGPS's performance. Interestingly, models that learn edge weights via attention mechanisms outperform MagNet on the directed GRN graph. This is likely because MagNet's

unitary evolution of complex-valued features does not resemble the actual feature propagation (i.e., the GRN dynamics), in addition to the shortcomings highlighted in the previous section. CoED outperforms the attention-based models in both datasets as it optimizes the edge directions directly as a part of the learning objective independent of the node features. The shortcomings of GAT compared to CoED on graph ensemble data are discussed further in Appendix H.

We then investigated whether CoED can recover the ground truth continuous edge directions of the triangular lattice graph, given that the feature propagation steps during data generation closely resemble the message-passing operation of CoED. As shown in Figure 4, CoED correctly learns the true directions. Lastly, since both synthetic datasets are generated by propagating input features over multiple hops, we investigated how performance scales with model depth by training CoED and the second-best model with up to 10 layers. Figure 5 demonstrates that CoED continues to improve as depth increases, while the performance of the other models plateau at a shallower depth.

|  | **Lattice** | **GRN** |
|---|---|---|
| **GCN** | 77.56 ±0.47 | 69.38 ±0.62 |
| **GAT** | 9.41 ±0.05 | 12.07 ±1.50 |
| **GraphGPS** | 3.47 ±0.14 | 25.16 ±1.56 |
| **MagNet** | 75.06 ±0.03 | 43.42 ±4.34 |
| **Chung** | 8.03 ±0.03 | 62.95±0.78 |
| **DRew** | 28.55 ±0.02 | 69.92 ±0.15 |
| **FLODE** | 7.54±0.05 | 70.31 ±0.03 |
| **CoED** | **1.36 ±0.06** | **5.02 ±0.45** |

Table 2: Comparison of different models on the synthetic datasets. Values are test losses reported with a common factor of $10^{-3}$ in both columns.

Figure 4: Learned theta vs. true theta for CoED applied to directed flow on triangular lattice synthetic dataset.

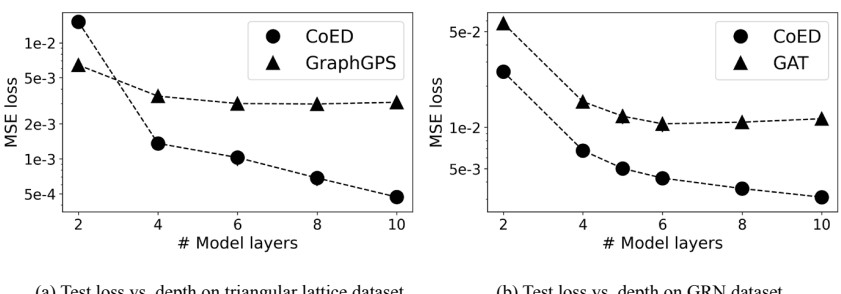

(a) Test loss vs. depth on triangular lattice dataset

(b) Test loss vs. depth on GRN dataset

Figure 5: Model performance as a function of depth.

### 5.2.2 REAL DATASETS

**Single-cell Perturb-seq.** Perturb-seq (Dixit et al., 2016) is a well-established experimental technique in single-cell biology that inspired the synthetic GRN experiment described earlier. In Perturb-seq experiments, one or more genes in a cell are knocked out resulting in zero expression—as in our synthetic GRN dataset. The resulting changes in the expression levels of all other genes are then measured to elucidate gene-gene interactions. For our study, we used the Replogle-gwps dataset (Replogle et al., 2022; Peidli et al., 2024), which includes 9,867 distinct single-gene perturbations, along with control measurements from cells without any perturbation to establish baseline gene expression levels. The learning task is again predicting the expression levels of all genes following perturbation given the initial steady-state with the expression levels of the perturbed genes set to zero. Since there is no ground truth gene regulatory network (GRN) available for this dataset, we constructed an undirected $k$-nearest neighbors graph to connect genes with highly correlated expression levels. All models are trained using this heuristic graph. Details of the data processing procedure are provided in Appendix A.3.1.

**Wikipedia web traffic.** We also modeled the traffic flow between Wikipedia articles using the WikiMath dataset, which is classified as a "static graph with temporal signals" in the PyTorch Ge-

ometric Temporal library (Rozemberczki et al., 2021). In this dataset, each node corresponds to a Wikipedia article on a popular mathematics topic, and each directed edge represents a link from one article to another. The node features are the daily visit counts of all articles over a period of 731 consecutive days. The learning task is node regression: predict the next day's visit counts across all articles given today's visit counts. We trained the baseline models using the ground truth directed graph, while CoED was trained starting from the undirected version of the graph. Additional details are provided in Appendix A.3.2.

**Power grid.** We applied CoED to the optimal power flow (OPF) problem using the OPF-Data (Lovett et al., 2024) from the PyTorch Geometric library. In this dataset, a power grid is represented as a directed graph with nodes corresponding to buses (connection points for generators and loads) and edges representing transformers and AC lines. Input features are the operating values of all components under specific load conditions, and the targets are the corresponding AC-OPF solution values at the generator nodes. To compare different models, we used a consistent architecture across components but substituted different model layers for message passing. For CoED, they were again converted to undirected edges. Additional details are provided in Appendix A.3.3.

**Results.** Table 3 reports the test performances of all baseline models and CoED on the three datasets. The baselines include GAT, MagNet, DirGCN, and DirGAT. We focus on these models because attention-based approaches showed competitive performance on the synthetic datasets, and MagNet, which accounts for edge directions, performed well on the directed GRN dataset. Details of the training setup, hyperparameter search procedure, and selected hyperparameters are provided in Appendix A.3.

|  | Perturb-seq | Web traffic | Power grid |
|---|---|---|---|
| GCN | 4.13±0.08 | 7.07±0.03 | 28.56±6.08 |
| MagNet | 4.11±0.01 | 6.94±0.02 | 18.05±2.77 |
| GAT | 3.85±0.03 | 6.00±0.03 | 13.57±1.73 |
| DirGCN | 5.46±0.26 | 6.72±0.04 | 6.15±0.84 |
| DirGAT | 3.98±0.07 | 6.55±0.04 | 3.28±0.17 |
| **CoED** | **3.56±0.03** | **5.76±0.05** | **2.91±0.11** |

Table 3: Comparison of different methods on real graph ensemble datasets. Values are test losses reported with common factors of $10^1, 10^{-1}, 10^{-3}$ for Perturb-seq, web traffic, and power grid columns, respectively.

We observe that CoED achieves the best performance across all three datasets. On the Perturb-seq dataset with an undirected graph, MagNet performs similarly to GCN while DirGCN struggles. We attribute DirGCN's poor performance to clashing learnable parameters: it uses two distinct weight matrices, $\mathbf{W}_{\leftarrow}$ and $\mathbf{W}_{\rightarrow}$, applied to identical in- and out-neighbor aggregated messages in the case of undirected graphs. For a propagation path of $L$ hops, this results in $2^L$ feature transformations, comprised of different combinations of the two weight matrices, which together reduce the model's ability to efficiently learn the optimal weight matrices. In contrast, DirGAT's attention mechanisms break the symmetry of the undirected edges, leading to improved performance. CoED naturally addresses this issue by learning the edge directions, which are visualized in Appendix I.1. On the web traffic and power grid datasets, which have directed graphs, we observe a similar trend. MagNet outperforms GCN due to its ability to process directed messages. However, GAT delivers better performance than MagNet, likely because its attention mechanism effectively captures important features. Since directed graphs create distinct feature propagation paths, DirGCN achieves substantial performance gains. DirGAT further improves upon DirGCN by leveraging additional edge weight learning through an attention mechanism. CoED surpasses all these models, demonstrating the effectiveness of learning continuous edge directions.

## 6 CONCLUSION

We have introduced the Continuous Edge Direction (CoED) GNN, which assigns fuzzy, continuous directions to the edges of a graph and employs a novel complex-valued Laplacian to transform information propagation on graphs from diffusion to directional flow. Our theoretical analysis shows that CoED GNN is more expressive than existing Laplacian-based methods and matches the expressiveness of an extended Weisfeiler-Leman (WL) test for directed graphs with fuzzy edges. Through extensive experiments on both synthetic and real-world graph ensemble datasets—including gene regulatory networks, web traffic, and power grids—we demonstrated that learning continuous edge directions significantly improves performance over existing GNN models.

ACKNOWLEDGEMENT

The authors thank Shishir Adhikari for his contributions to the conceptual formulation of this work and for numerous insightful discussions on graph Laplacians on directed graphs. We acknowledge funding from the National Institutes of Health (NIH)–National Heart, Lung, and Blood Institute (NHLBI)–under grant R01HL158269, and Advanced Research Projects Agency for Health (ARPA-H) award 1AYSAX000005.

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

# A EXPERIMENTAL DETAILS

## A.1 NODE CLASSIFICATION

In Table 1, MLP, GCN, SAGE, GAT, GGCN, and Sheaf's results on Texas, Wisconsin, Citeseer, and Cora are taken from Bodnar et al. (2022); MLP, GCN, MagNet, and DirGNN's results on Roman-Empire, SNAP-Patents, and Arxiv-Year from Rossi et al. (2024); SAGE and GAT's results on Roman-Empire, and GCN, SAGE, and GAT's results on the filtered versions of Squirrel and Chameleon from Platonov et al. (2023); GCN, SAGE, GAT, and GGCN's results on AM-Computers and AM-Photo, and GGCN's result on Roman-Empire from Deng et al. (2024), SAGE and GAT's results on SNAP-Patents from Dwivedi et al. (2023); MagNet's result on Texas and Wisconsin from the original paper (Zhang et al., 2021) and the filtered versions of Squirrel and Chameleon from (Sun et al., 2024); Flode's results on Roman-Empire, Citeseer, and Cora from the original paper (Maskey et al., 2023); Co-GNN's results on Roman-Empire and Cora from the original paper (Finkelshtein et al., 2023); GAT's result on Arxiv-Year from Lim et al. (2021). We trained CoED and baseline models to fill the remaining entries in the table. We describe the training procedures below. Texas, Wisconsin, Citeseer, and Cora datasets were downloaded using PyTorch Geometric library (Fey & Lenssen, 2019) with `split='geom-gcn'` argument to use the 10 fixed 48%/32%/20% training/validation/test splits provided by (Pei et al., 2020). We downloaded AM-Computers and AM-Photo datasets from the same library but used the 60%/20%/20%-split file provided in the repository of Deng et al. (2024). Since the original Squirrel and Chameleon datasets (Pei et al., 2020) have redundant nodes, we used the filtered versions with directed graphs provided in the repository of Platonov et al. (2023). For Roman-Empire, SNAP-Patents, and Arxiv-Year datasets, we used the dataloading pipeline provided in the repository of Rossi et al. (2024). All referenced results used the same splits as in our experiments.

**Training.** We evaluated the validation accuracy at each epoch, incrementing a counter if the value did not improve and resetting it to 0 when a new best validation accuracy was achieved. Training was early-stopped when the counter reached a patience of 200. Unless otherwise mentioned, we used the default hyperparameter settings of the respective models. We used the ReLU activation function and the ADAM optimizer in all experiments. Across all models, we searched over the following hyperparmeters: hidden dimension $\in [16, 256]$, learning rate $\in [$5e-4, 2e-2$]$, weight decay $\in [0, $1e-2$]$, dropout rate $\in [0, 0.7]$, and the number of layers $\in [2, 5]$. We additionally searched over model-specific hyperparameters: the weight between in-/out-neighbor aggregated messages $\alpha \in \{0, 0.5, 1\}$, jumping knowledge (jk) $\in \{$None, 'cat', 'max'$\}$, and layer-wise feature normalization (norm) $\in \{$True, False$\}$ for DirGNN, Chung, and CoED; self-feature transform $\in \{$True, False$\}$, self-loop value $\in \{0, 1\}$ for Chung and CoED; convolution type $\in \{$'GCN', 'SAGE', 'GAT'$\}$ for DirGNN; the order of Chebyshev polynomial $K \in \{1, 2\}$, the global directionality $q \in [0, 0.25]$, and self-loop value $\in \{0, 1\}$ for MagNet; the initial temperature for Gumbel-softmax $\tau_0 \in \{0, 0.1\}$, the number of environment network layers $\in [1, 4]$, the hidden dimension of environment network $\in [16, 128]$, the number of action network layers $\in \{1, 2\}$, the hidden dimension of action network $\{4, 8\}$, layer norm $\in \{$True, False$\}$, skip connection $\in \{$True, False$\}$, model type in $\in \{$'Sum GNN', 'Mean GNN', 'GCN'$\}$; and the number of layers $\in [1, 3]$, the number of both encoder and decoder layers $\in \{1, 2\}$, and self-loop value $\in \{0, 1\}$ for FLODE. For FLODE, the number of layers refers to the number of forward Euler steps, and we solved the heat equation with a minus sign, which was the default setup for node classification tasks. If the self-loop value is 1, the self-feature is combined with neighbors' features in the AGGREGATE function. For CoED and Chung model on Roman-Empire, SNAP-Patents, and Arxiv-Year datasets, we only searched over the type of jumping knowledge $\in \{$'cat', 'max'$\}$, setting all other hyperparameters as reported in Rossi et al. (2024). For Sheaf, we searched over stalk dimension $\in [3, 8]$, hidden dimension $\in [8, 64]$, the number of layers $\in [2, 8]$, sheaf weight decay from the same weight decay range, the number of decoder layer $\in \{1, 2\}$, and using both low-pass and high-pass filters $\in \{$True, False$\}$. We report in Table 1 the mean accuracy and standard deviation over the 10 test splits using the best hyperparameters presented below. All experiments were performed on two NVIDIA RTX 6000 Ada Generation GPUs with 48GB of memory and one NVIDIA A100 Tensor Core GPU with 80GB, and it took roughly three weeks of training to produce the results. We report OOM when a model with the minimum hyperparameter configuration fails to process data on a 48GB GPU.

| | # layers | # hidden | lr | wd | dropout | self-loop | $\alpha$ | norm | jk | self-feature |
|---|---|---|---|---|---|---|---|---|---|---|
| Roman-Empire | 5 | 256 | 1e-2 | 0 | 0.2 | 0 | 0.5 | False | max | True |
| SNAP-Patents | 5 | 32 | 1e-2 | 0 | 0 | 0 | 0.5 | True | cat | True |
| Texas | 2 | 64 | 2e-2 | 5e-4 | 0.5 | 0 | 0.5 | False | None | True |
| Wisconsin | 2 | 128 | 2e-2 | 1e-3 | 0.5 | 0 | 0.5 | False | None | True |
| Arxiv-Year | 6 | 256 | 5e-3 | 0 | 0 | 0 | 0.5 | False | max | False |
| Squirrel | 3 | 64 | 1e-2 | 5e-3 | 0 | 0 | 0.5 | False | cat | False |
| Chameleon | 2 | 64 | 1e-2 | 2e-3 | 0 | 0 | 0.5 | False | cat | False |
| Citeseer | 2 | 256 | 2e-3 | 0 | 0.7 | 0 | 0.5 | False | None | True |
| AM-Computers | 2 | 512 | 5e-3 | 0 | 0.7 | 1 | 0 | False | None | False |
| AM-Photo | 3 | 128 | 1e-3 | 0 | 0.7 | 1 | 0.5 | False | None | True |
| Cora | 2 | 128 | 5e-4 | 1e-4 | 0.5 | 1 | 0 | False | None | False |

Table A.1: Selected hyperparameters CoED.

| | # env/act layers | # env/act hidden | $\tau_0$ | conv type | lr | wd | dropout | layer norm | skip |
|---|---|---|---|---|---|---|---|---|---|
| SNAP-Patents | 2 / 1 | 32 / 4 | 0 | GCN | 1e-2 | 0 | 0 | True | False |
| Texas | 3 / 1 | 64 / 4 | 0.1 | GCN | 2e-2 | 1e-3 | 0.5 | False | True |
| Wisconsin | 4 / 2 | 64 / 4 | 0.1 | GCN | 2e-2 | 5e-4 | 0.5 | False | True |
| Arxiv-Year | 3 / 1 | 128 / 8 | 0.1 | GCN | 1e-3 | 0 | 0 | True | True |
| Squirrel | 2 / 1 | 64 / 4 | 0.1 | Mean GCN | 5e-3 | 0 | 0 | False | False |
| Chameleon | 2 / 1 | 64 / 4 | 0.1 | Mean GCN | 5e-3 | 0 | 0 | False | False |
| Citeseer | 4 / 1 | 64 / 4 | 0 | GCN | 1e-2 | 0 | 0.7 | True | True |
| AM-Computers | 4 / 2 | 64 / 4 | 0 | GCN | 5e-3 | 0 | 0 | True | False |
| AM-Photo | 3 / 2 | 64 / 8 | 0 | GCN | 5e-3 | 0 | 0.5 | True | True |

Table A.2: Selected Hyperparameters for Co-GNN.

| | # layers | # hidden | conv type | lr | wd | dropout | $\alpha$ | norm | jk |
|---|---|---|---|---|---|---|---|---|---|
| Texas | 2 | 256 | DirSAGE | 2e-2 | 5e-4 | 0.5 | 0.5 | False | None |
| Wisconsin | 3 | 256 | DirSAGE | 1e-2 | 1e-4 | 0.5 | 0.5 | False | None |
| Squirrel | 4 | 64 | DirGCN | 1e-2 | 5e-3 | 0 | 0.5 | False | cat |
| Chameleon | 2 | 64 | DirGCN | 1e-2 | 2e-3 | 0 | 0.5 | False | cat |
| Citeseer | 2 | 128 | DirSAGE | 5e-3 | 1e-3 | 0.5 | 0.5 | False | None |
| AM-Computers | 3 | 256 | DirSAGE | 1e-3 | 0 | 0.5 | 0.5 | True | None |
| AM-Photo | 4 | 128 | DirSAGE | 1e-3 | 0 | 0.5 | 0.5 | True | None |
| Cora | 2 | 64 | DirGCN | 5e-3 | 5e-4 | 0.5 | 0 | False | None |

Table A.3: Selected hyperparameters for DirGNN.

| | # layers | # hidden | lr | wd | dropout | $\alpha$ | norm | jk | self-feature |
|---|---|---|---|---|---|---|---|---|---|
| Roman-Empire | 5 | 256 | 1e-2 | 0 | 0.2 | 0.5 | False | cat | True |
| SNAP-Patents | 5 | 32 | 1e-2 | 0 | 0 | 0.5 | True | cat | True |
| Texas | 3 | 64 | 1e-2 | 1e-3 | 0 | 0.5 | True | None | True |
| Wisconsin | 2 | 32 | 1e-2 | 1e-4 | 0.5 | 0.5 | False | None | True |
| Arxiv-Year | 6 | 256 | 5e-3 | 0 | 0 | 0.5 | False | cat | False |
| Squirrel | 2 | 32 | 5e-3 | 5e-3 | 0 | 0.5 | False | cat | False |
| Chameleon | 2 | 128 | 2e-2 | 1e-3 | 0 | 0.5 | False | cat | False |
| Citeseer | 2 | 64 | 1e-3 | 1e-4 | 0.5 | 0 | False | None | True |
| AM-Computers | 3 | 256 | 1e-3 | 0 | 0 | 0 | True | None | True |
| AM-Photo | 4 | 256 | 5e-3 | 1e-4 | 0 | 0 | True | None | True |
| Cora | 2 | 32 | 5e-4 | 0 | 0.5 | 0 | False | None | True |

Table A.4: Selected hyperparameters for Chung.

| | # layers | # hidden | lr | wd | dropout | K | q |
|---|---|---|---|---|---|---|---|
| Citeseer | 3 | 256 | 1e-3 | 0 | 0.5 | 2 | 0 |
| AM-Computers | 5 | 256 | 1e-3 | 1e-4 | 0.5 | 2 | 0 |
| AM-Photo | 5 | 256 | 2e-3 | 0 | 0.7 | 3 | 0 |
| Cora | 1 | 32 | 1e-2 | 0 | 0.5 | 2 | 0 |

Table A.5: Selected hyperparameters for MagNet.

| | # layers | # hidden | type | # stalk | lr | wd | sheaf wd | dropout | high/low-pass filter | # encoder layers |
|---|---|---|---|---|---|---|---|---|---|---|
| Roman-Empire | 6 | 16 | Diagonal | 3 | 2e-3 | 1e-2 | 0 | 0.7 | False/True | 2 |
| Arxiv-Year | 8 | 16 | Diagonal | 4 | 1e-2 | 0 | 0 | 0 | True/False | 1 |
| Squirrel | 2 | 8 | Orthogonal | 6 | 1e-3 | 0 | 0 | 0 | Ture/False | 1 |
| Chameleon | 2 | 16 | Diagonal | 6 | 1e-2 | 0 | 0 | 0 | True/False | 1 |
| AM-Computers | 4 | 32 | General | 6 | 1e-2 | 0 | 0 | 0 | True/False | 1 |
| AM-Photo | 4 | 16 | Diagonal | 8 | 1e-2 | 0 | 0 | 0 | True/False | 1 |

Table A.6: Selected hyperparameters for Sheaf.

| | # layers | # hidden | # encoder/decoder layers | lr | wd | dropout | self-loop |
|---|---|---|---|---|---|---|---|
| Texas | 1 | 128 | 1 / 1 | 1e-2 | 0 | 0 | 0 |
| Wisconsin | 1 | 128 | 1 / 1 | 5e-3 | 0 | 0.5 | 0 |
| Squirrel | 1 | 16 | 1 / 1 | 2e-2 | 1e-2 | 0.5 | 1 |
| Chameleon | 1 | 256 | 1 / 1 | 1e-2 | 5e-3 | 0.5 | 1 |
| AM-Computers | 1 | 32 | 1 / 2 | 5e-3 | 0 | 0 | 1 |
| AM-Photo | 1 | 64 | 2 / 1 | 5e-3 | 5e-3 | 0.5 | 1 |

Table A.7: Selected hyperparameters for FLODE.

| | # layers | # hidden | lr | wd | dropout |
|---|---|---|---|---|---|
| MLP (Squirrel) | 3 | 64 | 1e-3 | 1e-4 | 0.5 |
| MLP (Chameleon) | 4 | 128 | 1e-2 | 0 | 0.7 |
| MLP (AM-Computers) | 2 | 256 | 5e-3 | 1e-4 | 0 |
| MLP (AM-Photo) | 2 | 256 | 5e-3 | 1e-4 | 0 |
| SAGE (Arxiv-Year) | 4 | 128 | 1e-3 | 1e-4 | 0 |
| GGCN (Squirrel) | 2 | 32 | 5e-3 | 1e-4 | 0.7 |
| GGCN (Chameleon) | 4 | 64 | 1e-2 | 5e-3 | 0.7 |

Table A.8: Selected hyperparameters for MLP, SAGE, and GGCN.

## A.2 GRAPH ENSEMBLE EXPERIMENT WITH SYNTHETIC DATASETS

### A.2.1 DATA GENERATION AND TRAINING SETUP FOR THE DIRECTED FLOW TRIANGULAR LATTICE GRAPH

**Data generation.** To obtain the triangular lattice graph described in the main text, we first designed a potential function $V$ on $[-2, 2]^2$ plane with a peak (source) and a valley (sink). We used quadratic potentials, located at $\mu_1 = (-1, 1)$ and $\mu_2 = (1, -1)$ with stiffness matrices,

$$\mathbf{K}_1 = \mathbf{K}_2 = \begin{pmatrix} 1 & 0 \\ 0 & 1 \end{pmatrix}$$

With magnitudes $a_1 = 1$ and $a_2 = -1$, the potential function $V(\mathbf{x})$ is parameterized as,

$$V(\mathbf{x}) = a_1(\mathbf{x} - \mu_1)^\top \mathbf{K}_1(\mathbf{x} - \mu_1) + a_2(\mathbf{x} - \mu_2)^\top \mathbf{K}_2(\mathbf{x} - \mu_2)$$

We then generated a triangular lattice on the 2d plane and considered this lattice as a graph $\mathcal{G}_{\text{lattice}}$, where the vertices of the lattice serve as the nodes of the graph, and the edges of the lattice form the edges of the graph. In this way, each node $v_i \in \mathcal{V}_{\text{lattice}}$ has an associated spatial position $\mathbf{x}_i$ on the 2d plane. We computed $\Delta V_{ij} = V(\mathbf{x}_j) - V(\mathbf{x}_i)$ for all $(v_i, v_j) \in \mathcal{E}_{\text{lattice}}$. All $\Delta V_{ij}$ values were shifted

and scaled to the range $[0, \pi/2]$ to obtain the $\theta_{ij}$, which is an approximate version of the gradient direction of the potential. In the resulting lattice graph, an edge points towards the node with the lower potential energy.

We then assigned to each node a 10-dimensional random feature vector sampled independently from the standard multivariate normal distribution, and normalized them to have a unit-norm, and repeated the process 500 times to generate an ensemble of node features. To generate corresponding target values, we propagated features using Equation 4 with message-passing matrices $\mathbf{P}_{\rightarrow}$ and $\mathbf{P}_{\leftarrow}$ computed from $\Theta$ of the lattice graph as described in Equation 2 and the entries of the $10 \times 10$ weight matrices $\mathbf{W}_{\rightarrow}$, $\mathbf{W}_{\leftarrow}$, and $\mathbf{W}_{\text{self}}$ sampled independently from the standard normal distribution and shared across all 10 iterations. Instead of applying an activation function, we normalized the features $\mathbf{m}_{\text{self}} + \mathbf{m}_{\rightarrow} + \mathbf{m}_{\leftarrow}$ to have unit norm. We used the features after 10 iterations of message passing as the target values. We divided these 500 instances of feature-target pairs using 60%/20%/20% random training/validation/test split.

**Training.** We used a batch size of 16 for training with random shuffling at each epoch and a full batch for both validation and testing. We evaluated the validation MSE at each epoch, incrementing a counter if the value did not improve and resetting it to 0 when a new best validation MSE was achieved. Training was early-stopped when the counter reached a patience of 20. We used neither dropout nor weight decay, as we aim to learn an exact mapping from node features to target values for regression, as opposed to a noise-robust node embedding for classification. We used the ReLU activation function in all models, except for ELU in GAT, and used ADAM optimizer for all experiments. Across all models, we searched over the following hyperparmeters: the number of layers $\in [2, 4]$, hidden dimension $\in [16, 64]$, learning rate $\in [\text{1e-3}, \text{1e-2}]$. We additionally grid-searched over model-specific hyperparameters: the number of attention heads $\in \{1, 4, 8\}$ and skip connection (sc) $\in \{\texttt{True}, \texttt{False}\}$ for GAT; attention type $\in \{\text{'multihead'}, \text{'performer'}\}$, attention heads $\in \{1, 4, 8\}$, encoding type $\in \{\text{'eigenvector'}, \text{'electrostatic'}\}$, the dimension of eigenvector encoding $\in [2, 5, 10, 20]$ and self-loop value $\in \{0, 1\}$ for GraphGPS; the order of Chebyshev polynomial $K \in \{1, 2\}$ and self-loop value $\in \{0, 1\}$ for MagNet; $\alpha \in \{0, 0.5, 0.1\}$ for Chung; multi-hop aggregation mechanism $\in \{\text{'sum'}, \text{'weight'}\}$ for DRew; the number of both encoder and decoder MLP layers $\in \{1, 2, 3\}$, and self-loop value $\in \{0, 1\}$ for FLODE; self-feature transform $\in \{\texttt{True}, \texttt{False}\}$, learning rate for $\Theta \in [\text{1e-3}, \text{1e-2}]$, and layer-wise (lw) $\Theta$ learning $\in \{\texttt{True}, \texttt{False}\}$ for CoED. For GraphGPS, we used GINE as a convolution layer and provided 1 as an edge attribute. Computing structural encoding via random walk resulted in an all-zero vector since the degree of node is 3 across all nodes except at the boundary in our lattice graph. We thus opted to use the electrostatic function encoding provided in the original paper as an alternative to structural encoding. For MagNet, we optimized $q$ along with the model parameters during training. We supplied the undirected version of the lattice graph by setting all $\theta_{ij}$ values to $\pi/4$ and used self-feature transform with self-loop value set to 0. We did not use layer-wise feature normalization. Table 2 reports the mean accuracy and standard deviation on the test data from the top 5 out of 7 training runs with different initializations using the best hyperparameters shown below. All experiments were performed on two NVIDIA RTX 6000 Ada Generation GPUs with 48GB of memory and it took about 3 days of training time to generate the results.

| Model | # layers | # hidden | lr | self-loop | sc | # attn. heads | attn. type | enc. type | K | $\alpha$ | aggr. | # enc. layers | # dec. layer | self-feature | lr $\Theta$ | lw $\Theta$ |
|---|---|---|---|---|---|---|---|---|---|---|---|---|---|---|---|---|
| GCN | 2 | 16 | 1e-3 | - | - | - | - | - | - | - | - | - | - | - | - | - |
| GAT | 4 | 32 | 1e-3 | - | True | 4 | - | - | - | - | - | - | - | - | - | - |
| GraphGPS | 4 | 64 | 1e-3 | 0 | - | 4 | multihead | eigenvector (dim=5) | - | - | - | - | - | - | - | - |
| MagNet | 4 | 64 | 1e-3 | 1 | - | - | - | - | 2 | - | - | - | - | - | - | - |
| Chung | 4 | 64 | 1e-3 | 0 | - | - | - | - | - | 0.5 | - | - | - | - | - | - |
| DRew | 3 | 64 | 5e-3 | - | - | - | - | - | - | - | weight | - | - | - | - | - |
| FLODE | 4 | 64 | 1e-2 | 0 | - | - | - | - | - | - | - | 1 | 3 | - | - | - |
| CoED | 4 | 64 | 1e-3 | - | - | - | - | - | - | - | - | - | - | True | 1e-3 | False |

Table A.9: Hyperparameters selected for node regression on the synthetic lattice graph.

### A.2.2 DATA GENERATION AND TRAINING SETUP FOR THE GRN DYNAMICS EXPERIMENT

**Data generation.** We prepared a directed adjacency matrix $\mathbf{A}$ for a graph with 200 nodes by sampling each entry of the matrix independently from a Bernoulli distribution with success probability 0.03. We interpreted an edge $A_{ij}$ as indicating that the gene represented by the node $v_j$ is regulating the gene represented by the node $v_i$. We then randomly chose half of the edges as activating edges and the other half as suppressing edges. The scalar feature value of each node is the expression level

of a gene, measured as concentration $c_i$. In order to simulate the gene regulatory network dynamics where genes are either up- or down- regulating connected genes, we sampled the magnitudes of activation $\gamma_{ij}^{\text{act}}$ and suppression $\gamma_{ij}^{\text{sup}}$ from a uniform distribution with the support $[0.5, 1.5]$. We additionally sampled half-saturation constants $K_{ij}$, which control how quickly $c_i$ changes in response to $c_j$, from a uniform distribution with support $[0.25, 0.75]$. Lastly, we sampled the initial concentrations independently from a uniform distribution with support $[0.1, 10]$ for each gene, and ran the GRN dynamics as described by,

$$\frac{dc_i}{dt} = \sum_{j \in \mathcal{N}(i)} \left( \gamma_{ij}^{\text{act}} F^{\text{act}}(c_j, K_{ij}) + \gamma_{ij}^{\text{sup}} F^{\text{sup}}(c_j, K_{ij}) \right) - c_i \tag{5}$$

for 250 time steps with $dt = 0.05$ to reach a steady state for $\mathbf{c}$. The summation in the above equation is over all genes that either activate or repress gene $i$. $F^{\text{act}}(c_j, K_{ij}) = c_j^2/(K_{ij}^2 + c_j^2)$ and $F^{\text{sup}}(c_j, K_{ij}) = K_{ij}^2/(K_{ij}^2 + c_j^2)$, which are the Hill functions defining up- and down- regulations of gene $i$ by gene $j$, respectively. From this steady state, we mimicked the gene knockout experiment in biology by setting the concentration values of a chosen set of genes to zero and running the GRN dynamics for additional 100 time steps, by which time genes reached new steady-state values. We performed a single-gene knockout for all 200 genes and a double-gene knockout for 1000 randomly selected pairs of genes. We defined node features as the original steady state with the values of knockout genes set to zero and the corresponding target values as the new steady-state values reached from this state. This procedure generates an ensemble of 1200 feature-target pairs for each node for the synthetic GRN graph. We used all 200 single gene knockout results as training data, and randomly selected 200 and 800 double gene knockout results for validation and testing, respectively.

**Training.** For each knockout result, we used all nodes for regression except those corresponding to the knocked out genes. We used a batch size of 8 for training with random shuffling at each epoch, and a full batch for both validation and testing. We evaluated the validation loss at every epoch, and implemented the same counting scheme as in the directed flow experiment to early-stop the training with a patience of 50. We searched over the number of layers $\in [2, 5]$, hidden dimension $\in \{16, 32\}$, and learning rate $\in [5\text{e-}4, 5\text{e-}3]$, and otherwise conducted the same hyperparameter search as described in the lattice experiment, using the same training setup. Table 2 reports the mean accuracy and standard deviation on the test data from the top 5 out of 7 training runs with different initializations using the best hyperparameters shown below. All experiments were performed on two NVIDIA RTX 6000 Ada Generation GPUs with 48GB of memory and it took about 3 days of training time to generate the results.

| Model | # layers | # hidden | lr | self-loop | sc | # attn. heads | attn. type | enc. type | K | $\alpha$ | aggr. | # enc. layers | # dec. layer | self-feature | lr $\Theta$ | lw $\Theta$ |
|---|---|---|---|---|---|---|---|---|---|---|---|---|---|---|---|---|
| GCN | 3 | 32 | 5e-4 | - | - | - | - | - | - | - | - | - | - | - | - | - |
| GAT | 5 | 32 | 2e-3 | - | True | 8 | - | - | - | - | - | - | - | - | - | - |
| GraphGPS | 5 | 32 | 5e-3 | 0 | - | 4 | multihead | eigenvector (dim=10) | - | - | - | - | - | - | - | - |
| MagNet | 5 | 32 | 5e-3 | 1 | - | - | - | - | 2 | - | - | - | - | - | - | - |
| Chung | 5 | 32 | 1e-3 | - | - | - | - | - | - | 0.5 | - | - | - | True | - | - |
| DRew | 3 | 32 | 5e-4 | - | - | - | - | - | - | - | weight | - | - | - | - | - |
| FLODE | 5 | 32 | 5e-3 | 1 | - | - | - | - | - | - | - | 1 | 1 | - | - | - |
| CoED | 5 | 32 | 1e-3 | - | - | - | - | - | - | - | - | - | - | True | 1e-2 | True |

Table A.10: Hyperparameters selected for node regression on the synthetic GRN graph.

## A.3 GRAPH ENSEMBLE EXPERIMENT WITH REAL DATASETS

### A.3.1 PREPROCESSING, DATA GENERATION, AND TRAINING SETUP FOR SINGLE-CELL PERTURBATION EXPERIMENTS

**Preprocessing.** We downloaded Replogle-gwps dataset from scPerturb database (Peidli et al., 2024) and followed the standard single-cell preprocessing routine using Scanpy software (Wolf et al., 2018), selecting for the top 2000 most variable genes. This involved running the following four functions: `filter_cells` function with `min_counts=20000` argument, `normalize_per_cell` function, `filter_genes` function with `min_cells=50` argument, `highly_variable_genes` function with `n_top_genes=2000`, `flavor='seurat_v3'`, and `layer='counts'` arguments. Afterwards, we discarded genes that were not part of the top 2000 most variable genes. These 2000 genes define nodes. We did not log transform the expression values and used the normalized expression values obtained from these preprocessing steps for all downstream tasks. Out of 9867 genes that were perturbed in the original dataset, 958 of them

were among the top 2000 most variable genes. Note that perturbed genes should have near zero expression values since the type of perturbation in the original experiment was gene knockout via a technique called CRISPRi. Therefore, we disregarded perturbed genes if the perturbations did not result in more than 50% of cells with zero expression values for each respective gene. This preprocessing step identifies 824 effective gene perturbations. Note that there are multiple measurements per perturbation.

**Data generation for node regression.**    In perturbation experiments, the expression values are measured 'post-perturbation' (equivalent to $c_i + \Delta c_i$ in Figure 3(d) of the main text). Consequently, we do not have access to their 'pre-interaction' expression levels (equivalent to $c_i$ with a perturbed gene's value set to 0). To pair each post-perturbation expression values to its putative pre-interaction state, we randomly sampled a control measurement and set the expression value of the perturbed gene to 0. These pairs of pre-interaction and post-perturbation expressions define the ensemble of features and targets. Since the number of measurements vary per perturbation, we standardized the dataset diversity by downsampling the feature target pairs to 2 per perturbation. We split each dataset based on perturbations, grouping all cells subject to the same perturbation together. We performed 60/10/30 training/validation/test splits and computed $k$-nearest neighbors gene-gene graph with $k = 3$ using the training split. We checked that this graph roughly corresponds to creating an edge between genes whose Pearson correlation coefficient is higher than 0.5. We also confirmed that this graph represents a single connected component.

**Training.**    As in the GRN example, we used all nodes (i.e., genes) for regression except for those corresponding to a knocked-out gene. We used a batch size of 16 for training with random shuffling at each epoch, and a full batch for both validation and testing. We evaluated the validation loss at every epoch, and implemented the same counting scheme as in the synthetic dataset experiments to early-stop the training with a patience of 30. Since the models' performances generally improved as their depths and hidden dimensions increased, we used 4 layers and a hidden dimension of 32 across all models to streamline the comparison. We searched over learning rate $\in \{$1e-3, 5e-3, 1e-2$\}$ for all models, the order of Chebyshev polynomial $K \in \{1, 2\}$ for MagNet; the number of attention heads $\in \{1, 4, 8\}$ for both GAT and DirGAT; additionally, skip connection (sc) $\in \{$True, False$\}$ for GAT; self-feature transform $\in \{$True, False$\}$, learning rate for $\Theta \in \{$5e-4, 1e-3, 5e-3, 1e-2$\}$ and layer-wise $\Theta$ learning $\in \{$True, False$\}$ for CoED. We learned $q$ in MagNet. Table 3 reports the mean accuracy and standard deviation on the test data from the top 5 out of 7 training runs with different initializations using the best hyperparameters shown below. All experiments were performed on two NVIDIA RTX 6000 Ada Generation GPUs with 48GB memory and approximately one day of training time was spent for generating the results.

| Model | lr | K | sc | # attn. heads | self-feature | lr $\Theta$ | lw $\Theta$ |
|---|---|---|---|---|---|---|---|
| GCN | 5e-3 | - | | - | - | - | - |
| MagNet | 1e-3 | 2 | | - | - | - | - |
| GAT | 1e-3 | - | True | 4 | - | - | - |
| DirGCN | 5e-3 | - | | - | - | - | - |
| DirGAT | 1e-3 | - | | 1 | - | - | - |
| CoED | 1e-3 | - | | - | False | 5e-4 | False |

Table A.11: Hyperparameters selected for node regression on the Perturb-seq data.

A.3.2    TRAINING SETUP FOR WEB TRAFFIC EXPERIMENTS

We downloaded WikiMath dataset from PyTorch Geometric Temporal library (Rozemberczki et al., 2021) with the default time lag value of 8. We followed the same temporal split from the paper where 90% of the snapshots were used for training and 10% of the forecasting horizons were used for testing.

**Training.**    The input features and target values are shaped as $N \times 8$ and $N \times 1$, respectively. Since we consider a pair of the two consecutive snapshots as input and the corresponding target, we disregarded the first 7 snapshots (i.e., feature dimensions) and used only the node values (i.e., visit counts of Wikipedia articles) of the last snapshot for prediction for testing. For training, we utilized all 8 snapshots, generating 8 predictions. MSE loss for the predictions from the first 7 snapshots were each evaluated with the next 7 snapshots as target values. This procedure is similar

to the 'incremental training mode' used to train time-series based models (Rozemberczki et al., 2021; Eliasof et al., 2024; Guan et al., 2022). We used a full batch for both training and testing. We evaluated test loss at each step, and implemented the same counting scheme used above to early-stop training with a patience value of 50. We used 2 layers with a hidden dimension of 16 across all models. We searched over learning rate $\in \{$1e-3, 5e-3, 1e-2, 2e-2$\}$ for all models and also over the same model-specific hyperparameters discussed in the Perturb-seq experiments except the number of attention heads, which was limited to 2 due to memory constraints. We learned $q$ for MagNet. Table 3 reports the mean accuracy and standard deviation on the test data from the top 5 out of 7 training runs with different initializations using the best hyperparameters shown below. All experiments were performed on two NVIDIA RTX 6000 Ada Generation GPUs with 48GB memory and approximately 16 hours of training time was spent for generating the results.

| Model | lr | K | sc | # attn. heads | self-feature | lr $\Theta$ | lw $\Theta$ |
|---|---|---|---|---|---|---|---|
| GCN | 1e-2 | - | | - | - | - | - |
| MagNet | 5e-3 | 1 | | - | - | - | - |
| GAT | 1e-2 | - | True | 2 | - | - | - |
| DirGCN | 2e-2 | - | | - | - | - | - |
| DirGAT | 5e-2 | - | | 1 | - | - | - |
| CoED | 5e-3 | - | | - | True | 1e-2 | False |

Table A.12: Hyperparameters selected for node regression on the WikiMath data.

### A.3.3 TRAINING SETUP FOR POWER GRID EXPERIMENTS

We downloaded a power grid graph with 2000 nodes from PyTorch Geometric library, compiled via OPFData (Lovett et al., 2024). We selected the 'fulltop' topology option to obtain the information of the entire graph, opposed to 'N-1' perturbation option which masks parts of the graphs. We randomly sampled 300 graphs and split them into 200 training set, 50 validation set, and 50 test set. Power grids are heterogeneous graphs. Refer to Figure 1 of Piloto et al. (2024) for detailed overview of the different components. Importantly, generators, loads, and shunts ('subnodes') are all connected to buses ('nodes') via edges, and buses are connected to one another via two types of edges, transformers and AC lines. Thus, the load profile across the graph informs the generator subnodes via the bus-to-bus edges. We describe the architectural design choices that we made to accomplish this task while facilitating effective model comparison.

**Model.** The primary goal is to ensure information flow into those buses connected to generators. We thus substituted different model layers to process messages over the two types of bus-to-bus edges while keeping the rest of the architecture unchanged. The processing steps are:

1. Transform node/subnode features and edge features using an MLP.
2. For each bus node, integrate the features of its subnodes into its own features via Graph-Conv (i.e., $\mathbf{W}_1 \mathbf{x}_i + \mathbf{W}_2 \sum_{j \in \mathcal{N}(i)} e_{ji} \cdot \mathbf{x}_j$ ).
3. Incorporate edge features into node features using GINE.
4. Iterate message-passing among bus nodes
5. Decode the features of bus nodes into generator operating point values.

We varied the message-passing mechanism in step 4 by applying the different Aggregate and Update functions of each of the models that we analyzed.

**Training.** Following the example training routine outlined in Lovett et al. (2024), we trained models to predict generator active and reactive power outputs and evaluated MSE loss against those values in the AC-OPF solutions. We did not incorporate AC-OPF constraints, as the focus of the experiments was to compare the message-passing capabilities of CoED with other models. We refer to Böttcher et al. (2023); Piloto et al. (2024) for predicting AC-OPF solutions that satisfy constraints. We used a batch size of 16 during training with random shuffling applied at each epoch. We evaluated the validation loss at every epoch and early-stopped the training with the same counting scheme with a patience of 50. We iterated the step 4 above 3 times (i.e., 3 layers) and used 32 hidden dimension. We searched over learning rate $\in \{$5e-4, 1e-3, 2e-3, 5e-3$\}$, and otherwise the same set of

hyperparameters considered in the Perturb-seq experiments. We learned $q$ for MagNet. Table 3 reports the mean accuracy and standard deviation on the test data from the top 5 out of 7 training runs with different initializations using the best hyperparameters shown below. All training were performed on two NVIDIA RTX 6000 Ada Generation GPUs with 48GB memory and approximately one day of training time was spent for generating the results.

| Model | lr | K | sc | # attn. heads | self-feature | lr $\Theta$ | lw $\Theta$ |
|---|---|---|---|---|---|---|---|
| GCN | 5e-3 | - | | - | - | - | - |
| MagNet | 1e-3 | 2 | | - | - | - | - |
| GAT | 2e-3 | - | False | 4 | - | - | - |
| DirGCN | 1e-3 | - | | - | - | - | - |
| DirGAT | 1e-3 | - | | 8 | - | - | - |
| CoED | 5e-4 | - | | - | False | 5e-3 | True |

Table A.13: Hyperparameters selected for node regression on the AC-OPF data.

## B  TIME AND SPACE COMPLEXITY

Let $V$ and $E$ denote the numbers of nodes and edges, and let $H$ denote hidden dimension, which we assume stays constant for two consecutive layers. At minimum, the message-passing mechanism outlined in 2 results in the time complexity that scales as $\mathcal{O}(EH + VH^2)$, where the first term corresponds to aggregating features with dimension $H$ and the second term stems from the matrix-matrix multiplication of node features and a weight matrix. The space complexity is $\mathcal{O}(E + H^2)$ due to the edges and the weight matrix.

With the sparse form of the phase matrix $\Theta$, CoED incurs an additional $\mathcal{O}(E)$ term both in the time complexity from computing fuzzy propagation matrices $\mathbf{P}_{\leftarrow/\rightarrow}$ and in the space complexity due to storing $\Theta$. Unless layer-wise $\Theta$ learning is employed, this computation happens once and thus only adds minimal overhead. The layer-wise $\Theta$ learning adds an $\mathcal{O}(EL)$ term to the total time and space complexities over $L$ layers. For comparison, however, we note that computing $S$-head attentions incurs an $\mathcal{O}(EH)$ term (or even $\mathcal{O}(EHS)$ term if feature dimension is not divided by $S$) in the time complexity and an $\mathcal{O}(ES)$ term in the space complexity per layer.

## C  POSITIONAL ENCODING USING THE FUZZY LAPLACIAN

Graph Laplacians can be used to assign a positional encoding to each node of a graph based on the connectivity patterns of the nodes of the graph. Using the fuzzy Laplacian, we can extend positional encoding to include variations in directions of the edges surrounding a node in addition to the connectivity pattern of the graph. To demonstrate the utility of the eigenvectors of the fuzzy Laplacian for positional encoding, we visualize the eigenvectors of the Laplacian computed from the triangular lattice graph (which has a trivial connectivity pattern as shown above by the random walk structural encoding being trivially zero) supplemented with two different sets of edge directions. In the first case, we obtained edge directions from the gradient of source-sink potential function described in A.2.1. These edge directions are visualized in Figure 3(b) of the main text. In the second case, we obtained the edge directions for the same triangular lattice graph, from the following solenoidal vector field,

$$F(\mathbf{x}) = \big(\sin(\pi x)\cos(\pi y), -\cos(\pi x)\sin(\pi y)\big)$$

which does not have sources or sinks but instead features a cyclic flow. To assign edge directions using the solenoidal vector field, we computed $\theta_{ij}$ as the angle between the unit vector pointing from $v_i$ to $v_j$ and the vector evaluated at the midpoint of an edge, i.e., $F\big((\mathbf{x}_2 - \mathbf{x}_1)/2\big)$. All $\theta_{ij}$ were scaled to range from 0 to $\pi/2$. As shown in Figure C.1(a-e), the real part of the eigenvector distinguishes the peak and valley from the region in between, whereas the phase of the eigenvector distinguish the peak from the valley. Similarly in Figure C.1(f-j), the real part of the eigenvector highlights the regions adjacent to the cyclic flows. The magnitude of the eigenvector picks out the centers of the four solenoids in the middle. Taken together, the eigenvectors of the fuzzy Laplacian contain positional information based on the directions of the edges as demonstrated here for the same graph but with different edge directions.

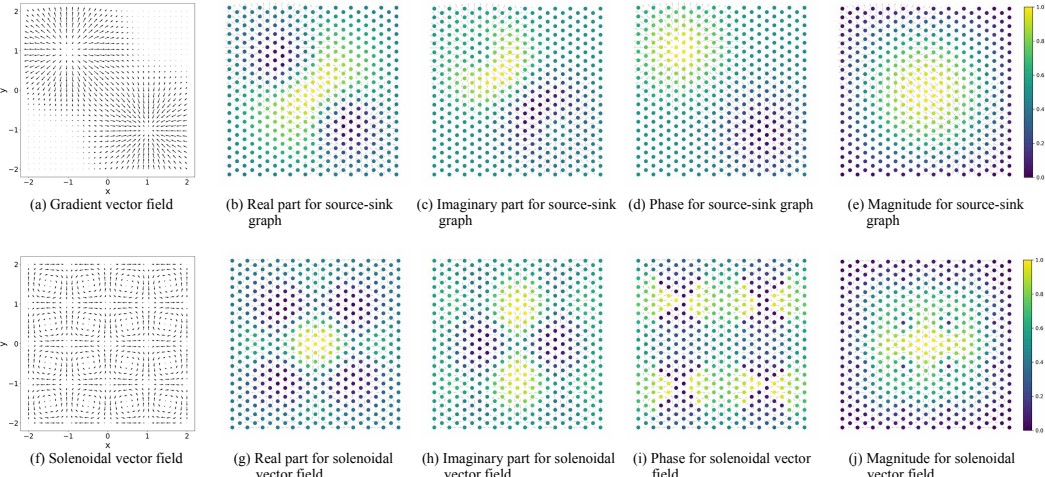

Figure C.1: Visualization of the eigenvector, corresponding to the eigenvalue with largest magnitude, of the fuzzy Laplacian of the triangular lattice whose edge directions are taken from the source-sink potential function described in the main text (top row) and the solenoidal vector field described in this section (bottom row). The original vector fields are shown in the left-most figures. The real and imaginary components, as well as the magnitude and phase, of the eigenvector encode positional information at each node about the direction of the edges surrounding that node.

# D MATHEMATICAL PROPERTIES OF THE FUZZY LAPLACIAN

We propose a new graph Laplacian matrix for directed graphs which generalizes to the case of directed graphs with fuzzy edges, where an edge connecting two nodes $A$ and $B$ can take on any intermediate value between the two extremes of pointing from $A$ to $B$ and pointing from $B$ to $A$. We will show that our Laplacian exhibits two useful properties: 1. The eigenvectors of our Laplacian matrix are orthogonal and therefore can be used as the positional encodings of the nodes of the graph. 2. For any node of the graph, our Laplacian aggregates information from neighbors that send information to the node separately from the neighbors that receive information from the node and is therefore as expressive as a weak form of the Weisfeiler-Leman (WL) graph isomorphism test for directed graphs with fuzzy edges that we define below.

## D.1 DIRECTED GRAPHS WITH FUZZY EDGES

We define a directed graph with fuzzy edges as follows. Our definition builds on the standard definition of a graph.

A *graph* $\mathcal{G}$ is an ordered pair $\mathcal{G} := (\mathcal{V}, \mathcal{E})$ comprising a set $\mathcal{V}$ of vertices or nodes together with a set $\mathcal{E}$ of edges. Each edge is a 2-element subset of $\mathcal{V}$.

- $\mathcal{V}$: A finite, non-empty set of vertices

$$\mathcal{V} = \{v_1, v_2, \ldots, v_n\}$$

- $\mathcal{E}$: A set of edges, each linking two vertices in $\mathcal{V}$

$$\mathcal{E} = \{(v_i, v_j) \mid v_i, v_j \in \mathcal{V}\}$$

To incorporate fuzzy directions to the edges, we define a new attribute for each edge.

- $\mu$: A function defining the direction of each edge

$$\mu : (v_i, v_j) \to [0, 1]$$

such that for each edge $(v_i, v_j) \in E$, $\mu(v_i, v_j) = x$ implies $\mu(v_j, v_i) = \sqrt{1 - x^2}$.

In this model, each edge is associated with a scalar $x$ that represents its direction. The value $x$ is a real number in the interval $[0, 1]$. If for an edge $(v_i, v_j)$, $\mu(v_i, v_j) = x$, then it must hold that $\mu(v_j, v_i) = \sqrt{1 - x^2}$, capturing the edge in both directions.

For example, if $\mu(v_i, v_j) = 1$ then the edge is an arc (directed edge) connecting node $v_i$ to node $v_j$. If $\mu(v_i, v_j) = 0$ then the edge is an arc connecting node $v_j$ to node $v_i$. If $\mu(v_i, v_j) = 1/\sqrt{2}$ then the edge is a bidirectional edge connecting node $v_i$ to node $v_j$ and node $v_j$ to node $v_i$.

For a scalar-directed edge graph $\mathcal{G} = (\mathcal{V}, \mathcal{E}, \mu)$, the adjacency matrix $\mathbf{A}$ is a square matrix of dimension $|\mathcal{V}| \times |\mathcal{V}|$. The entry $A_{ij}$ of the matrix is defined as follows:

$$
A_{ij} = \begin{cases} \mu(v_i, v_j), & \text{if } (v_i, v_j) \in E \\ 0, & \text{otherwise} \end{cases}
$$

In this setting, $\mu(v_i, v_j)$ captures the direction of the edge from $v_i$ to $v_j$. It follows that if an edge is present between nodes $v_i$ and $v_j$ then $A_{ji} = \sqrt{1 - A_{ij}^2}$.

Therefore, the adjacency matrix captures not only the presence of edges but also their direction according to the function $\mu$.

## D.2   Fuzzy Laplacian Matrix

For a scalar-directed edge graph $\mathcal{G} = (\mathcal{V}, \mathcal{E}, \mu)$ with adjacency matrix $\mathbf{A}$, we define its Fuzzy Laplacian matrix $\mathbf{L}_F$ as follows:

The diagonal entries of $\mathbf{L}_F$ are zero:
$$
(\mathbf{L}_F)_{ii} = 0
$$

The off-diagonal elements are

$$
(\mathbf{L}_F)_{ij} = \begin{cases} 0 & \text{if } A_{ij} = A_{ji} = 0 \\ e^{i\theta_{ij}} & \text{otherwise} \end{cases} \tag{6}
$$

where $\theta_{ij}$ is selected such that:
$$
\cos(\theta_{ij}) = A_{ij}
$$

In other words, the real part of $e^{i\theta_{ij}}$ is equal to the corresponding adjacency matrix entry $A_{ij}$. We require that $0 \leq \theta_{ij} \leq \pi/2$. It follows that $\theta_{ji} = \pi/2 - \theta_{ij}$.

The Fuzzy Laplacian $\mathbf{L}_F$ satisfies the property:
$$
\mathbf{L}_F = i\mathbf{L}_F^*
$$

To confirm this, note that $e^{-i\theta_{ij}} = \cos(-\theta_{ij}) + i\sin(-\theta_{ij}) = \cos(\theta_{ij}) - i\sin(\theta_{ij})$. Therefore, $(\mathbf{L}_F)_{ji} = \sin(\theta_{ij}) + i\cos(\theta_{ij}) = ie^{-i\theta_{ij}}$, and thus $\mathbf{L}_F = i\mathbf{L}_F^*$.

The fuzzy Laplacian takes the following form,

$$
\mathbf{L}_F = \begin{pmatrix} 0 & \cdots & & \cdots \\ \vdots & \ddots & & e^{i\theta_{ij}} \\ \vdots & ie^{-i\theta_{ij}} & & \ddots \end{pmatrix} \tag{7}
$$

Here, $e^{i\theta_{ij}}$ and $ie^{-i\theta_{ij}}$ are sample off-diagonal elements corresponding to the edge $(v_i, v_j)$ in the graph.

## D.3   Properties of Fuzzy Laplacian Matrix $\mathbf{L}_F$

In this section, we will show that the Fuzzy Laplacian matrix $\mathbf{L}_F$ has eigenvalues of the form $a + ia$, where $a \in \mathbb{R}$, and orthogonal eigenvectors.

### D.3.1 Eigenvalues of the Form $a + ia$

$\mathbf{L}_F$ has eigenvalues of the form $a + ia$ with $a \in \mathbb{R}$.

*Proof:*

Let $\lambda$ be an eigenvalue of $\mathbf{L}_F$, and let $\mathbf{w}$ be the corresponding eigenvector. Then:

$$\mathbf{L}_F \mathbf{w} = \lambda \mathbf{w} \;\; \Rightarrow \;\; \mathbf{w}^* \mathbf{L}_F^* = \lambda^* \mathbf{w}^* \;\; \Rightarrow \;\; \mathbf{w}^* \mathbf{L}_F^* \mathbf{w} = \lambda^* \mathbf{w}^* \mathbf{w}$$

$$\Rightarrow \; -i\mathbf{w}^* \mathbf{L}_F \mathbf{w} = \lambda^* \mathbf{w}^* \mathbf{w} \;\; \Rightarrow \;\; -i\lambda \mathbf{w}^* \mathbf{w} = \lambda^* \mathbf{w}^* \mathbf{w} \;\; \Rightarrow \;\; -i\lambda = \lambda^*$$

where we used $\mathbf{L}_F^* = -i\mathbf{L}_F$ to go to the second line. The last identity holds only when $\lambda = a + ia$ where $a$ is a real number.

### D.3.2 Orthogonal Eigenvectors

To prove that $\mathbf{L}_F$ has orthogonal eigenvectors, we need to show that if $\mathbf{w}$ and $\mathbf{v}$ are eigenvectors corresponding to distinct eigenvalues $\lambda_1$ and $\lambda_2$ respectively, then $\mathbf{w}$ and $\mathbf{v}$ are orthogonal.

*Proof:*

Let $\mathbf{L}_F \mathbf{w} = \lambda_1 \mathbf{w}$ and $\mathbf{L}_F \mathbf{v} = \lambda_2 \mathbf{v}$.

$$\mathbf{v}^* \mathbf{L}_F \mathbf{w} = \lambda_1 \mathbf{v}^* \mathbf{w} \;\; \Rightarrow \;\; i\mathbf{v}^* \mathbf{L}_F^* \mathbf{w} = \lambda_1 \mathbf{v}^* \mathbf{w} \;\; \Rightarrow \;\; i\lambda_2^* \mathbf{v}^* \mathbf{w} = \lambda_1 \mathbf{v}^* \mathbf{w} \;\; \Rightarrow \;\; \lambda_2 \mathbf{v}^* \mathbf{w} = \lambda_1 \mathbf{v}^* \mathbf{w}$$

where we used $\lambda = i\lambda^*$ derived above for the last step. Therefore, $\lambda_2 \mathbf{v}^* \mathbf{w} = \lambda_1 \mathbf{v}^* \mathbf{w}$. Since $\lambda_1 \neq \lambda_2$, it must be that $\mathbf{v}^* \mathbf{w} = 0$, i.e., $\mathbf{w}$ and $\mathbf{v}$ are orthogonal.

Because the eigenvectors of the Fuzzy Laplacian matrix $\mathbf{L}_F$ are orthogonal, they can be used as the positional encoding of the nodes of the graph.

## E Expressivity of Neural Networks using the Fuzzy Laplacian

### E.1 Graph isomorphism for directed graphs with fuzzy edges

First, we extend the standard definition of graph isomorphism to the case of directed graphs with fuzzy edges following a similar approach as in Piperno et al. (2018).

We only consider graph with a finite number of nodes. Therefore, the set of all edge weights form a countable set with finite cardinality.

Two directed fuzzy graphs $\mathcal{G} = (\mathcal{V}_\mathcal{G}, \mathcal{E}_\mathcal{G}, \mu_\mathcal{G})$ and $\mathcal{H} = (\mathcal{V}_\mathcal{H}, \mathcal{E}_\mathcal{H}, \mu_\mathcal{H})$ are said to be isomorphic if there exists a bijection $f : \mathcal{V}_\mathcal{G} \to \mathcal{V}_\mathcal{H}$ such that, for every pair of vertices $u, v$ in $\mathcal{V}_\mathcal{G}$, the following conditions hold:

1. $(u, v)$ is an edge in $\mathcal{E}_\mathcal{G}$ if and only if $(f(u), f(v))$ is an edge in $\mathcal{E}_\mathcal{H}$ for all vertices $u$ and $v$ in $\mathcal{V}_\mathcal{G}$.

2. For every edge $(u, v)$ in $\mathcal{E}_\mathcal{G}$ that maps to edge $(f(u), f(v))$ in $\mathcal{E}_\mathcal{H}$, the corresponding weights satisfy:

$$\mu_\mathcal{G}(u, v) = \mu_\mathcal{H}(f(u), f(v))$$

### E.2 Weisfeiler-Leman test for isomorphism of directed graphs with fuzzy edges

Next, we extend the Weisfeiler-Leman (WL) graph isomorphism test to determine whether two directed graphs with fuzzy edges are isomorphic or not according to the extended definition of graph isomorphism stated above.

WL test is a vertex refinement algorithm that assigns starting features to each node of the graph. The algorithm then aggregates all the features of each node's neighbors and hashes the aggregated labels alongside the node's own label into a unique new label. At each iteration of the algorithm, the list of labels is compared across the two graphs. If the labels are different, the two graphs are not isomorphic. If the labels are no longer updated at each iteration, the two graphs can potentially be isomorphic.

The Weisfeiler-Lehman (WL) graph isomorphism test treats all edges emanating from a node equivalently, as long as they connect to neighbors with the same label. Neighbors are distinguished solely by their labels, not by the attributes of the edges that connect them. When extending the WL test to directed graphs with fuzzy edges, we must account for the fact that fuzzy edges disrupt this uniform treatment of edges. We introduce the following two extensions of the WL test to handle directed graphs with fuzzy edges.

- **Strong form**: A node can distinguish its neighbors not only by their labels but also by the weights of the edges connecting them. This allows the node to differentiate between neighbors of the same label based on edge attributes.

- **Weak form**: A node cannot differentiate between neighbors of the same label based on edge weights. Instead, the only weight-related information a node can use is the sum of the edge weights connecting it to all neighbors with a given label.

These extensions adapt the WL test to account for the additional complexity introduced by fuzzy edges in directed graphs. Importantly, for all the proofs that follow, we will assume that the graph has a finite number of nodes.

**The strong form of WL test for fuzzy directed graphs:**

Given a directed graph with fuzzy edges $\mathcal{G} = (\mathcal{V}, \mathcal{E}, \mu)$, the strong form of the WL test calculates a node coloring $C^{(t)} : \mathcal{V} \to \{1, 2, \ldots, k\}$, a surjective function that maps each vertex to a color. At the first iteration, $C^{(0)} = 0$. At all subsequent iterations,

$$C^{(t)}(i) = Relabel(C^{(t-1)}(i), \{\!\{(\mu_{ij}, C_j^{(t-1)}) : j \in N(i)\}\!\}), \tag{8}$$

where $(\mu_{ij}, C_j^{(t-1)})$ is tuple comprised of the weight of the edge from node $i$ to node $j$ and $C_j^{(t-1)}$, the color of node $j$ in the previous iteration. To simplify notation, we will write $\mu_{ij}$ to denote the weight of the edge connecting nodes $i$ and $j$ instead of $\mu(v_i, v_j)$ from now on. $N(i)$ denotes the set of all neighbors of node $i$. Function $Relabel$ is an injective function that assigns a unique new color to each node based on the node's color in the previous iteration and the tuples formed from its neighbors colors and the value of the edges connecting the node to those neighbors.

From the definition above, it follows that a graph neural network $\Gamma : \mathcal{G} \to \mathbb{R}^d$ that aggregates and updates the node features as follows

$$h_i^{(k)} = \phi(h_i^{(k-1)}, f(\{\!\{(\mu_{ij}, h_j^{(k-1)}) : j \in N(i)\}\!\})), \tag{9}$$

will map two graphs $\mathcal{G}_1$ and $\mathcal{G}_2$ to different vectors in $\mathbb{R}^d$ if the above strong form of the graph coloring test deems that the two graphs are not isomorphic. In the above equation, $h_v^k$ is the hidden representation (or feature) of node $i$ at the $k$th layer. $\phi$ and $f$ are injective functions with $f$ acting on multisets of tuples of the edge weights and hidden representation of the neighbors of node $i$.

The proof of above statement is a trivial extension of the proof of Theorem 3 in (Xu et al., 2019). Briefly, the multiset of the features of the neighbors of node $i$ can be converted from a multiset of tuples of the form $\{\!\{(\mu_{ij}, h_j^{(k-1)}) : j \in N(i)\}\!\}$ to a multiset of augmented features of the neighbors $\{\!\{\tilde{h}_j^{(k-1)} : j \in N(i)\}\!\}$ where $\tilde{h}_j^{(k-1)} = (\mu_{ij}, h_j^{(k-1)})$. Because $\mu_{ij}$ form a set of finite cardinality, the augmented feature space $\tilde{h}_j^{(k-1)}$ is still a countable set and the same proof as in (Xu et al., 2019) can be applied.

**The weak form of WL test for fuzzy directed graphs:**

We introduce a weaker form of the WL test for directed graphs with fuzzy edges where a given node cannot distinguish between its neighboring nodes that have the same color based on the weights of the edges that they share. Rather, the node aggregates the weights of all outgoing and incoming edges of its neighbors of a given color.

The weak form of the algorithm is as follows. At the first iteration, $C^{(0)} = 0$. At all subsequent iterations,

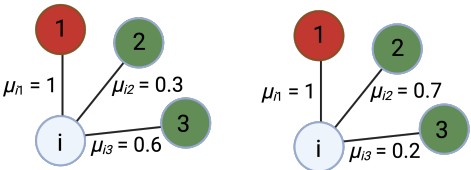

Figure E.1: Examples of two neighborhoods of a node $i$ are shown on the left and right. The strong form of the WL test for directed graphs with fuzzy edges can distinguish these two neighborhoods. The weak form of the WL test, however, cannot do so because the sum of the weights of the edges connecting node $i$ to green-colored neighbors is 0.9 in both cases.

$$
\begin{aligned}
C^{(t)}(i) = Relabel\Big( & C^{(t-1)}(i), \\
& \{( \sum_{j \in N(i)} \delta(C_j^{(t-1)} - c)\mu_{ij}, \\
& \sum_{j \in N(i)} \delta(C_j^{(t-1)} - c)\mu_{ji},\ c) : c \in C_{N(i)}^{(t-1)}\}\Big),
\end{aligned}
\tag{10}
$$

The term $\sum_{j \in N(i)} \delta(C_j^{(t-1)} - c)\mu_{ij}$ is summing over the edge weights $\mu_{ij}$ of all neighbors of $i$ that have color $c$. The tuple now also contains a similar that sums over the edge weights pointing from node $j$ to node $i$, $\mu_{ji}$ for all neighbors of the same color. As stated above, the edge weights are related, namely, $\mu_{ji} = \sqrt{1 - \mu_{ij}^2}$. The color of the node $i$ at the previous iteration alongside the set of tuples containing the sum of incoming edge weights from neighbors of a given color, the sum of outgoing edge weights to the neighbors of a given color, and the color of those neighbors are inputs to the injective function $Relabel$, which assigns a unique color to node $i$ for the next iteration.

**Theorem**. Let $\Gamma : \mathcal{G} \to \mathbb{R}^d$ be a graph neural network that updates node features as follows,

$$
h_i^{(k)} = MLP^{(k)}\left(h_i^{(k-1)}, \Re\left[\sum_{j \in N(i)} (\mathbf{L}_F)_{ij} h_j^{(k-1)}\right], \Im\left[\sum_{j \in N(i)} (\mathbf{L}_F)_{ij} h_j^{(k-1)}\right]\right),
\tag{11}
$$

where $MLP^{(k)}$ is a multi-layer perceptron. The last two terms of the MLP input are the real and imaginary parts of the aggregated features of the neighbors of node $i$, $\sum_{j \in N(i)}(\mathbf{L}_F)_{ij} h_j^{(k-1)}$. With sufficient number of layers, the parameters of $\Gamma$ can be learned such that it is as expressive as the weak form of the WL test, in that $\Gamma$ maps two graphs $\mathcal{G}_1$ and $\mathcal{G}_2$ that the weak form of WL test decides to be non-isomorphic to different embeddings in $\mathbb{R}^d$.

**Proof**. First, we claim that there exists a GNN of the form:

$$
h_i^{(k)} = \phi\left(f(h_i^{(k-1)}), \sum_{j \in N(i)} \mu_{ij} f(h_j^{(k-1)}), \sum_{j \in N(i)} \mu_{ji} f(h_j^{(k-1)})\right),
\tag{12}
$$

that is as expressive as the weak form of the WL test. We will prove this by induction.

Note that $F(i) = \sum_{j \in N(i)} \mu_{ij} f(h_j^{(k-1)})$ and $G(i) = \sum_{j \in N(i)} \mu_{ji} f(h_j^{(k-1)})$ are only injective up to the sum of the incoming and outgoing weights for a given type of neighbor (see Figure E.1 for an example). This is because the weak form of the WL test only accounts for the sum of the weights of outgoing and incoming edges to neighboring nodes of a given color. This is different from previous work that dealt with undirected graphs (Xu et al., 2019) or graphs with directed but unweighted edges (Rossi et al., 2024).

Let's denote with $X$ the multiset of the colors of all neighbors of node $i$ after a given number of iterations of the color refinement algorithm. Equivalently, we can consider the multiset of all the features of the neighboring nodes of $i$ after a given number of iteration of the GNN algorithm.

Because our graphs have a finite number of nodes, $X$ is bounded. For graphs without directed edges or weighted edges, it can be shown (see Lemma 5 of (Xu et al., 2019)) that there always exists an injective function $f$ such that $F(x) = \sum_{x \in X} f(x)$ is injective for all multisets $X$, i.e. if $F(X) = F(Y)$ for two multisets $X$ and $Y$ then $X = Y$. This result can be easily extended to regular directed graphs without weighted edges by considering the neighbors with incoming edges separately from the neighbors with outgoing edges (Rossi et al., 2024). For such graphs, a function $f$ exists such that the tuple $(\sum_{x \in X_\rightarrow} f(x), \sum_{x \in X_\leftarrow} f(x))$ is only the same for two nodes if they have identical neighborhoods. $X_\rightarrow$ and $X_\leftarrow$ denotes the multiset of features of the neighboring nodes that are connected to $i$ using an outgoing edge and incoming edge respectively.

Let's proceed with our proof by induction. At the first iteration, $k = 0$, all the nodes have the same color corresponding to the same trivial feature (e.g. scalar 0). Functions $F(i)$ and $G(i)$ then simply sum the weights of the outgoing and incoming edges of node $i$ respectively and use their values to assign updated feature $h_i^{(1)}$. This is identical to the procedure that the weak WL algorithm is using to assign new colors to the nodes at its first iteration. Therefore, nodes that would be assigned a given color under the WL color refinement algorithm at its first iteration will also be assigned the same updated feature vector $h_i^{(1)}$.

Assume that our claim hold for iteration $k - 1$. This mean that all the nodes that the weak WL color refinement assigns to different colors are also assigned different features $h_i^{(k-1)}$ by the GNN. Following the weak WL test, at iteration $k$, we need to sum the incoming and outgoing edges across all the neighbors with the same $h_j^{(k-1)}$. An example of an $f$ that allows this is one-hot encoding of all features. There always exists a number $N \in \mathbb{N}$ such that the features $h_i^{(k-1)}$ across all nodes $i$ of the graph can be one-hot encoded in an $N$ dimensional vector. It follows,

$$\sum_{j \in N(i)} \mu_{ij} f(h_j^{(k-1)}) = \left[ \sum_{j \in N(i)} \delta(h_j^{(k-1)} - h) \mu_{ij} \right]_{h \in H_{N(i)}^{(k-1)}}, \tag{13}$$

where $H_{N(i)}^{(k-1)}$ is the set of all the features $h_j^{(k-1)}$ of nodes $j$ that are neighbors of node $i$. A similar expression can be written for the sum over the incoming weights $\mu_{ji}$. Thus, at iteration $k$ the GNN will assign distinct features to all nodes that are also assigned a distinct color at iteration $k$ of the weak WL color refinement algorithm.

To complete the proof, we note that because $\mu_{ji} = \sqrt{1 - \mu_{ij}^2}$, we can reparameterize the edge weights as $\mu_{ij} = \cos(\theta_{ij})$ and $\mu_{ji} = \sin(\theta_{ij})$ with $0 \leq \theta_{ij} \leq \pi/2$. Eq. 12 can be rewritten as:

$$h_i^{(k)} = \phi \left( f(h_i^{(k-1)}), \Re \left[ \sum_{j \in N(i)} (\mathbf{L}_F)_{ij} f(h_j^{(k-1)}) \right], \Im \left[ \sum_{j \in N(i)} (\mathbf{L}_F)_{ij} f(h_j^{(k-1)}) \right] \right), \tag{14}$$

Finally, we use universal approximation theorem of multilayer perceptrons (Hornik et al., 1989) to model both the $\phi$ computation at layer $k$ and the $f$ computation for the next layer, $k + 1$. Namely $MLP^{(k)}$ denotes $f^{(k+1)} \circ \phi^{(k)}$. Taken together, the GNN in Eq. 11 is as expressive as the weak form of the WL color refinement algorithm for directed graphs with fuzzy edges.

# F  MAGNETIC LAPLACIAN IS NOT AS EXPRESSIVE AS THE FUZZY LAPLACIAN.

## F.1  MAGNETIC LAPLACIAN

A commonly used Laplacian for directed graphs is the Magnetic Laplacian. To construct the Magnetic Laplacian, we start with the asymmetric adjacency matrix of the graph and symmetrize it. Note that we are using the conventional definition of adjacency matrix in this section and not the fuzzy definition used above. The adjacency matrix $\mathbf{A}$ for a directed graph is a binary matrix, where

$A_{ij} = 1$ represents the presence of a directed edge from vertex $v_i$ to vertex $v_j$ and $A_{ij} = 0$ represents the absence of such an edge.

The symmetrized adjacency matrix $S$ is defined as:

$$S_{ij} = \frac{1}{2}(A_{ij} + A_{ji}).$$

To capture the direction of the edges, a phase matrix is defined as,

$$\Theta_{ij}^{(q)} = 2\pi q(A_{ij} - A_{ji}). \tag{15}$$

The Hermitian adjacency matrix is defined as the element-wise product of the above two matrices,

$$\mathbf{H}^{(q)} = S \odot \exp(\mathrm{i}\Theta^{(q)}).$$

$\mathbf{H}^{(q)}$ has some useful properties in capturing the directionality of the edges of the graph. For example, for $q = 1/4$, if there is an edge connecting $j$ to $k$ but no edge connecting $k$ to $j$ then $\mathbf{H}_{jk}^{(q)} = \mathrm{i}/2$ and $\mathbf{H}_{kj}^{(q)} = -\mathrm{i}/2$. Although this encoding of edge direction is useful, the fact that both incoming and outgoing edges are purely imaginary (only different up to a sign) means that in general it is impossible to distinguish features aggregated from neighbors that are connected to a node through outgoing edges from those connected through incoming edges. The fuzzy Laplacian $\mathbf{L}_F$, however, trivially distinguishes features from outgoing neighbors from those of incoming neighbors by keeping one set real and the other imaginary. We will expand on the implication of this for the expressivity of graph neural networks constructed using these two approaches below.

### F.2 Limitations of the Magnetic Laplacian

Conventionally, a magnetic Laplacian is defined by subtracting $\mathbf{H}^{(q)}$ from the degree matrix, $\mathbf{L}_M = \mathbf{D} - \mathbf{H}^{(q)}$. If such a Laplacian matrix is used to aggregate information of the nodes of the graph, the features of a node itself are combined with the features of its neighboring nodes. For directed graphs, there are two categories of neighboring nodes, those connected to a node $i$ with outgoing edges from $i$ ($A_{ij} = 1$ and $A_{ji} = 0$) and those connected to node $i$ with incoming edges ($A_{ji} = 1$ and $A_{ij} = 0$). Of course, it is possible for a neighboring node to be both an outgoing and incoming neighbor, in which case $A_{ji} = A_{ij} = 1$.

Features from these two categories of neighbors, alongside the feature of the node itself, form three distinct categories that in general must be kept distinct for maximum expressivity (Rossi et al., 2024). Using a complex Laplacian for aggregating the features across the nodes allows a simple mechanism for distinguishing two categories (corresponding to the real and imaginary parts of the resulting complex features) but not all three. To get around this limitation, we keep the self-feature of every node distinct from the aggregated features of its neighbors and concatenate it with the aggregated feature prior to applying the multi-layer perceptron to update the features from one layer to the next. Therefore, to maximize the expressivity of the magnetic Laplacian, we set the diagonal terms of the Laplacian matrix to zero, and define the magnetic Laplacian simply as $\mathbf{L}_M = \mathbf{H}^{(q)}$.

**Lemma.** For simple directed graphs without fuzzy edges, using the magnetic Laplacian $\mathbf{L}_M$ in the graph neural network of Eq. 11 in the place of the fuzzy Laplacian $\mathbf{L}_F$,

$$h_i^{(k)} = MLP^{(k)} \left( h_i^{(k-1)}, \Re \left[ \sum_{j \in N(i)} (\mathbf{L}_M)_{ij} h_j^{(k-1)} \right], \Im \left[ \sum_{j \in N(i)} (\mathbf{L}_M)_{ij} h_j^{(k-1)} \right] \right), \tag{16}$$

does not decrease the expressivity of the graph neural network in that both networks are as expressive as the weak form of the WL graph isomorphism test.

**Lemma.** To prove this statement, we will show that the MLP can learn a linear combination of its input that would it make it equivalent to the graph neural network defined in Eq. 11. This is because when $q = 1/8$, the real and imaginary parts of the aggregated features of the neighboring nodes are linear combinations of the features of the outgoing and incoming neighbors.

The fuzzy Laplacian $\mathbf{L}_F$ directly separates the features of outgoing and incoming neighbors in the real and imaginary components of the aggregated features respectively. Define $F_{\to}(i) =$

$\Re\left[\sum_{j\in N(i)}(\mathbf{L}_F)_{ij}h_j^{(k-1)}\right]$ and $F_{\leftarrow}(i) = \Im\left[\sum_{j\in N(i)}(\mathbf{L}_F)_{ij}h_j^{(k-1)}\right]$ as the features aggregated from the outgoing and incoming neighbors respectively. With $q = 1/8$, we have,

$$\Re\left[\sum_j(\mathbf{L}_M)_{ij}h_j^{(k-1)}\right] = \frac{1}{2\sqrt{2}}(F_{\rightarrow}(i) + F_{\leftarrow}(i))$$

$$\Im\left[\sum_j(\mathbf{L}_M)_{ij}h_j^{(k-1)}\right] = \frac{1}{2\sqrt{2}}(F_{\rightarrow}(i) - F_{\leftarrow}(i))$$

The MLP layer then in the graph neural network of Eq.16 simply needs to learn a linear combination of its second and third concatenated inputs to become equivalent to the graph neural network of Eq.11. Therefore, the two graph neural networks are equally as expressive.

Next, we extend the definition of the magnetic Laplacian to directed graphs with fuzzy edges, defined above, which have weighted edges that indicate intermediate values of directionality between the two extremes of the edge pointing form node $i$ to node $j$ and the edge pointing from node $j$ to node $i$. We implemented these intermediate values in the fuzzy Laplacian by assigning a weight $\mu_{ij}$ to each edge. Although not necessary in general, we further assumed that $\mu_{ji} = \sqrt{1 - \mu_{ij}^2}$. This constraint allowed us to think of each edge weight as an angle $\theta_{ij} = \arccos(\mu_{ij})$ which we then used to define the fuzzy Laplacian matrix (Eq.7).

To extend the magnetic Laplacian to directed graphs with fuzzy edges, we can assign a different value for $q$ (Eq. 15) to each edge. To keep the notation comparable to that of the fuzzy Laplacian, we instead assign a separate angle $\theta_{ij}$ to each edge and define the magnetic Laplacian as, $(\mathbf{L}_M)_{ij} = e^{i\theta_{ij}}$ for $j > i$ if there is an edge between the nodes $i$ and $j$, $S_{ij} \neq 0$. To ensure that magnetic Laplacian remains Hermitian, $\mathbf{L}_M^* = \mathbf{L}_M$, we require that $(\mathbf{L}_M)_{ji} = e^{-i\theta_{ij}}$, which defines the $\theta_{ij}$ values for $j < i$. It follows that $\theta_{ji} = -\theta_{ij}$.

$$\mathbf{L}_M = \begin{pmatrix} 0 & \cdots & & \cdots \\ \vdots & \ddots & & e^{i\theta_{ij}} \\ \vdots & e^{-i\theta_{ij}} & & \ddots \end{pmatrix}$$

It remains to be shown how we can assign the $\theta_{ij}$ values from the $\mu_{ij}$ in a self-consistent way.

**Theorem.** For directed graphs with fuzzy edges, the graph neural network constructed using the magnetic Laplacian $\mathbf{L}_M$, Eq. 16, is not as expressive as the weak form of the WL graph isomorphism test, and therefore not as expressive as the graph neural network defined using the fuzzy Laplacian $\mathbf{L}_F$, Eq. 11.

**Proof.** The Fuzzy Laplacian matrix conveniently captures the outgoing and incoming weights of the edges from one node to another, $(\mathbf{L}_F)_{ij} = \cos(\theta_{ij}) + i\sin(\theta_{ij})$. Namely, $\Re(\mathbf{L}_F)_{ij} = \cos(\theta_{ij})$ is the outgoing weight $\mu_{ij}$ of node $i$ to node $j$ and $\Im(\mathbf{L}_F)_{ij} = \sin(\theta_{ij})$ is in the incoming weight from node $j$ to node $i$, $\mu_{ji}$. Importantly, the outgoing weight from node $i$ to node $j$, $\mu_{ij}$, is the same as the incoming weight from node $i$ to node $j$. Similarly, the incoming weight from node $j$ to node $i$, $\mu_{ji}$ is the same as the outgoing weight from node $j$ to node $i$. These relationships are directly captured in the fuzzy Laplacian because $(\mathbf{L}_F)_{ji} = \sin(\theta_{ij}) + i\cos(\theta_{ij})$.

Let's consider how we can construct the magnetic Laplacian $\mathbf{L}_M$ from $\mu_{ij}$. The weight of the edge from $i$ to $j$ is $\mu_{ij}$. The weight from $j$ to $i$ is $\mu_{ij} = \sqrt{1 - \mu_{ij}^2}$.

From our definition above, $(\mathbf{L}_M)_{ij} = \cos(\theta_{ij}) + i\sin(\theta_{ij})$ and $(\mathbf{L}_M)_{ji} = \cos(\theta_{ij}) - i\sin(\theta_{ij})$. To relate $\theta_{ij}$ to $\mu_{ij}$ we note that the ratio of outgoing and incoming weights for node $i$ is $\mu_{ij}/\mu_{ji}$ and for node $j$ is $\mu_{ji}/\mu_{ij}$. Because these two quantities are reciprocals of each other, we can define

$$\ln\frac{\mu_{ij}}{\mu_{ji}} = \tan(2\theta_{ij}).$$

We chose the tan function on the right hand side because $\tan(-2\theta_{ij}) = -\tan(2\theta_{ij})$. Moreover, the log-ratio on the left hand side of above equation can take on any value from $-\infty$ to $\infty$. The right

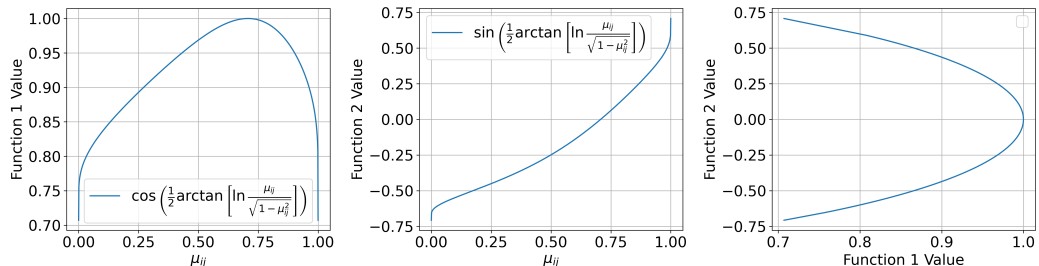

Figure F.1: The functions used to map the edge weights $\mu_{ij}$ to the $\theta_{ij}$ values of the magnetic Laplacian. $\cos(\theta_{ij})$ is plotted on the left. $\sin(\theta_{ij})$ is plotted in the middle. $\sin(\theta_{ij})$ versus $\cos(\theta_{ij})$ is plotted on the right.

hand side can take on a similar range of values if we allow $-\pi/4 \leq \theta_{ij} \leq \pi/4$. The specific form of this function does not matter as long it is an odd function that spans the specified domain and range.

Assume that each node of the graph has a trivial scalar feature equal to 1 (the first iteration of the WL test). The magnetic Laplacian applied to this graph to aggregate the features of the neighbors of node $i$ gives,

$$\Re\left[\sum_j (\mathbf{L}_M)_{ij}\right] = \sum_j \cos\left(\frac{1}{2}\arctan\left[\ln\frac{\mu_{ij}}{\sqrt{1-\mu_{ij}^2}}\right]\right)$$

$$\Im\left[\sum_j (\mathbf{L}_M)_{ij}\right] = \sum_j \sin\left(\frac{1}{2}\arctan\left[\ln\frac{\mu_{ij}}{\sqrt{1-\mu_{ij}^2}}\right]\right)$$

The above functions for a single value of $\mu_{ij}$ are plotted in Figure F.1. Importantly, these functions are not an injective function of all possible neighborhoods of node $i$. Consider the case of a node $i$ that has 5 neighbors with $\mu_{ij}$ values that result in $\cos(\theta_{ij}) = 0.8$ and $\sin(\theta_{ij}) = 0.6$, and another set of 5 neighbors with $\mu_{ij}$ values that result in $\cos(\theta_{ij}) = 0.8$ and $\sin(\theta_{ij}) = -0.6$. It follows that $\Re\left[\sum_j (\mathbf{L}_M)_{ij}\right] = 8$ and $\Im\left[\sum_j (\mathbf{L}_M)_{ij}\right] = 0$. The same values would have been obtained if node $i$ had a neighborhood comprised of 8 neighbors with $\mu_{ij}$ values such that $\cos(\theta_{ij}) = 1$ and $\sin(\theta_{ij}) = 0$. Therefore, the magnetic Laplacian cannot distinguish distinct neighborhoods and is not as expressive as the weak form of the WL test.

In practice, the more important limitation of the magnetic Laplacian is that it aggregates neighborhood information that results in linear combinations of the outgoing and incoming features to a node. In contrast, the fuzzy Laplacian by construct always keeps these contributions separate in the real and imaginary parts of the aggregated feature vector. In many applications, we would like to access the self-feature of the node and the incoming and outgoing aggregated features of its neighbors separately. The fuzzy Laplacian is thus a better choice. It could be argued that the MLP of a graph neural network can learn to disentangle the linear combinations of the outgoing and incoming features aggregated by the magnetic Laplacian. In general, however, this is not possible by the types of the graph neural networks that we considered here. This is because the parameters are MLP are the same for all the nodes of the graph. The linear combinations, however, depend on the specific weights $\mu_{ij}$ connecting node $i$ to its neighboring nodes $j$. Therefore, in general, the MLP will not be able to learn to disentangle the linear combinations of the incoming and outgoing features aggregated by the magnetic Laplacian.

## G   RUNTIME ANALYSIS

We conducted the runtime analysis to evaluate the computational efficiency of CoED in comparison to other representative baseline models. All models were configured with three layers and 64 hidden dimensions. We measured the forward and backward pass times on three graphs of increasing size.

Table G.1 reports the runtime of the models on datasets used for the node classification task, where edge directions are not learned. In this case, CoED's computational overhead stems from computing the in- and out-propagation matrices, $\mathbf{P}_{\leftarrow/\rightarrow}$, once at the initial layer. Consequently, its runtime is comparable to DirGCN, which employs a similar architecture.

Table G.2 presents the runtime for the batches of triangular lattice graph ensemble data. When including backpropagation for phase angles, CoED's backward pass time is approximately twice that of DirGCN. If unique edge directions are learned at each layer, CoED computes different propagation matrices at every layer. While this increases the forward pass time, the backward pass time remains unchanged since the backpropagation contributions from intermediate layers are subleading compared to the backpropagation to the initial layer.

| Dataset | GCN | | GAT | | MagNet | | DirGCN | | CoED | |
|---|---|---|---|---|---|---|---|---|---|---|
| | Forward | Backward | Forward | Backward | Forward | Backward | Forward | Backward | Forward | Backward |
| CORA | 4.8 | 8.3 | 25.3 | 62.9 | 89.3 | 120.6 | 14.3 | 17.6 | 14.1 | 21.9 |
| Roman-Empire | 16.9 | 37.2 | 218.9 | 646.1 | 300.1 | 331.5 | 30.0 | 48.7 | 34.5 | 66.1 |
| Arxiv-Year | 450.4 | 985.0 | 3379.2 | 10354.7 | 5058.2 | 7792.1 | 458.8 | 838.9 | 690.4 | 1231.4 |

Table G.1: Runtime of the forward and backward passes in ther node classification task. Times are reported in milliseconds. We set the number of layers to 3 and the hidden dimensions to 64 for all models. Note that we do not compute gradients with respect to phase angles $\Theta$ in this case. Cora, Roman-Empire, and Arxiv-Year respectively have 2708, 22662, and 169343 nodes and 10556, 44363, and 1166243 edges.

| Batch Size | GCN | | GAT | | MagNet | | GraphGPS | | DirGCN | | DirGAT | | CoED | | CoED (layerwise) | |
|---|---|---|---|---|---|---|---|---|---|---|---|---|---|---|---|---|
| | Forward | Backward | Forward | Backward | Forward | Backward | Forward | Backward | Forward | Backward | Forward | Backward | Forward | Backward | Forward | Backward |
| 4 | 2.5 | 5.0 | 15.4 | 35.2 | 14.4 | 22.5 | 14.1 | 31.3 | 4.4 | 6.1 | 26.1 | 59.7 | 5.9 | 11.0 | 6.8 | 12.4 |
| 32 | 11.0 | 24.6 | 123.3 | 362.8 | 81.1 | 128.7 | 103.6 | 285.5 | 20.9 | 24.1 | 226.4 | 754.8 | 20.4 | 50.0 | 44.5 | 52.8 |
| 256 | 105.0 | 254.4 | 1352.1 | 4644.2 | 964.2 | 1426.6 | 1101.0 | 3140.0 | 266.8 | 234.2 | 2261.2 | 9423.2 | 174.9 | 533.8 | 545.8 | 542.9 |

Table G.2: Runtime of the forward and backward passes in the node regression task with graph ensemble dataset. Times are reported in milliseconds. All models have 3 layers with 64 hidden dimensions. The backward pass of CoED now involves gradients with respect to the phase angles. At 4, 32, and 256 batch sizes, there are 1796, 14368, and 114944 nodes and 5048, 40384, and 323072 edges, respectively.

## H    Contrasting CoED GNN with Graph Attention Network (GAT)

Graph Attention Network (GAT) (Veličković et al., 2018), similar to CoED GNN, effectively learns edge weights prior to aggregating the features across neighboring nodes. Importantly, the learned edge weights are not necessarily symmetric. In the classic formulations of GAT, the attention weight from node $i$ to $j$ ($\alpha_{ij}$) depends on the concatenation of their feature vectors ($\mathbf{u}_i||\mathbf{u}_j$) and the shared learnable parameter $\mathbf{a}$: $e_{ij} = \text{LeakyReLU}(\mathbf{a}^T[\mathbf{u}_i||\mathbf{u}_j])$. Similarly, the attention weight from $j$ to $i$ ($\alpha_{ji}$) is computed as: $e_{ji} = \text{LeakyReLU}(\mathbf{a}^T[\mathbf{u}_j||\mathbf{u}_i])$. Since $\mathbf{u}_i||\mathbf{u}_j \neq \mathbf{u}_j||\mathbf{u}_i$, the raw attention scores $e_{ij}$ and $e_{ji}$ will generally differ. Even if a symmetric function is used to compute the attention weight, the attention coefficients $\alpha_{ij}$ are computed using a softmax function: $\alpha_{ij} = \frac{\exp(e_{ij})}{\sum_{k \in \mathcal{N}(i)} \exp(e_{ik})}$. This normalization is performed separately for the neighbors of $i$ and $j$, so even if $e_{ij}$ were equal to $e_{ji}$, the normalization step would generally make $\alpha_{ij} \neq \alpha_{ji}$. Therefore, it might appear that GATs can learn arbitrary asymmetric edge weights, much like CoED's continuous edge directions.

The key difference between CoED GNN and attention-based models is that CoED GNN does not rely on node features to learn continuous edge directions. Instead, the continuous edge directions are learned directly as part of optimizing for the given learning task. Consider the toy problem illustrated in Figure H.1. In this example, nodes in a linear-chain graph are assigned random input features that are independently and identically distributed. The output feature for each node is generated by shifting the input features along the chain in the clockwise direction by one hop. A training dataset consists of multiple realizations of the graph, each with different input and output node features, for a node regression task. GAT fails to correctly propagate the features across the graph in this setup because the node features are entirely uninformative. Consequently, it cannot learn meaningful edge

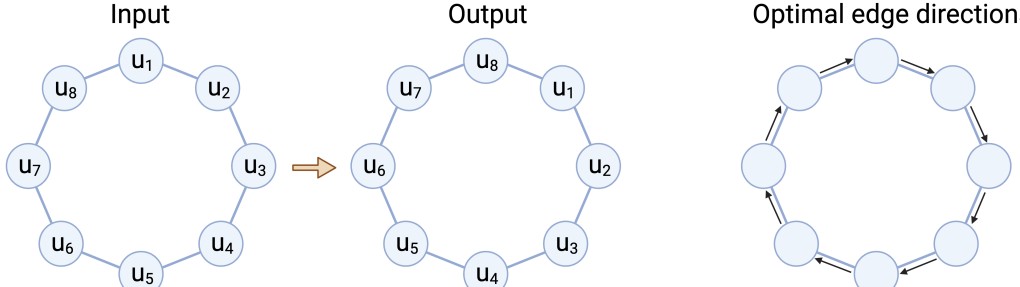

Figure H.1: A toy problem of learning to shift node features by one node in the clockwise direction for a circularized linear chain of nodes. In this hypothetical regression problem, the input are the original node features and the output are the node features shifted by one. The node features $\mathbf{u}_i$ are independently and identically distributed and therefore completely uninformative for determining the direction in which node features should be sent. The training data is an ensemble of the same linear-chain graph with different realizations of the node features. Any method that relies on node features to propagate the features (such as GAT) will fail at this task. CoED GNN, however, can learn the optimal edge directions for this task (shown on the right).

weights from the node features to perform the regression task. In contrast, CoED GNN successfully learns the optimal edge directions required to accomplish this task. The failure of GAT on directed graphs with random features has also been empirically demonstrated in the stochastic block model experiments in Zhang et al. (2021).

In many tasks, it is necessary to learn distinct edge directions in different parts of a graph, even when the node features are identical. The GAT framework cannot achieve this because it relies solely on node features to compute edge weights. In contrast, CoED directly learns continuous edge directions by optimizing them specifically for the task. To illustrate this, we compare GAT and CoED on a synthetic triangular lattice node regression task, as described in the main text, using node features derived from CIFAR images (Figure H.2). In this scenario, many nodes share nearly identical features, such as those representing blue sky pixels in an image. GAT fails to learn the correct edge directions under these conditions, while CoED successfully identifies the optimal directions for the task (Figure H.3).

Tables 2 and 3 in the main text empirically demonstrate that CoED GNN outperforms GAT on both synthetic and real-world datasets. The superior performance of CoED GNN may stem from the consistency of edge directions: the outgoing message from node $i$ to node $j$ must match the incoming message from node $j$ to node $i$. This consistency can be enforced in GAT by requiring that $\alpha_{ij} = 1 - \alpha_{ji}$. To test this hypothesis, we constructed a modified GAT model enforcing this requirement and applied it to the synthetic node regression task of directed flow on a triangular lattice graph, as described in the main text. To clarify the comparison, instead of using random features, we assigned CIFAR image pixel values as features for each node. Edge weights between nodes $i$ and $j$ were determined using the GAT formulation: $\alpha_{ij} = \mathrm{LeakyReLU}(\mathbf{a}^T[\mathbf{u}_i\|\mathbf{u}_j])$, where $\mathbf{a}$ is a learnable parameter. Importantly, we enforced the constraint $\alpha_{ji} = 1 - \alpha_{ij}$. As shown in Figure H.3, this modified GAT model, even with self-consistent edge weights, fails to learn the node regression task, much like the conventional GAT formulation. In contrast, CoED GNN successfully learns optimal edge directions and accurately predicts the output values.

Finally, we provide empirical evidence that CoED GNN can mitigate oversmoothing and facilitate long-range information transmission across graphs. In undirected graphs, information diffuses rather than flows because when node $i$ passes a message to node $j$, node $j$ simultaneously passes a message back to node $i$. This bidirectional exchange causes the diversity of node features to diminish rapidly as they converge to the averaged features across all nodes. Learning optimal edge directions addresses this issue by enabling directed information flow, preventing uniform diffusion (see Figure 1).

To demonstrate this empirically, we applied CoED GNN to the synthetic triangular lattice graph problem, where node features are derived from CIFAR image pixels as described earlier. Figure

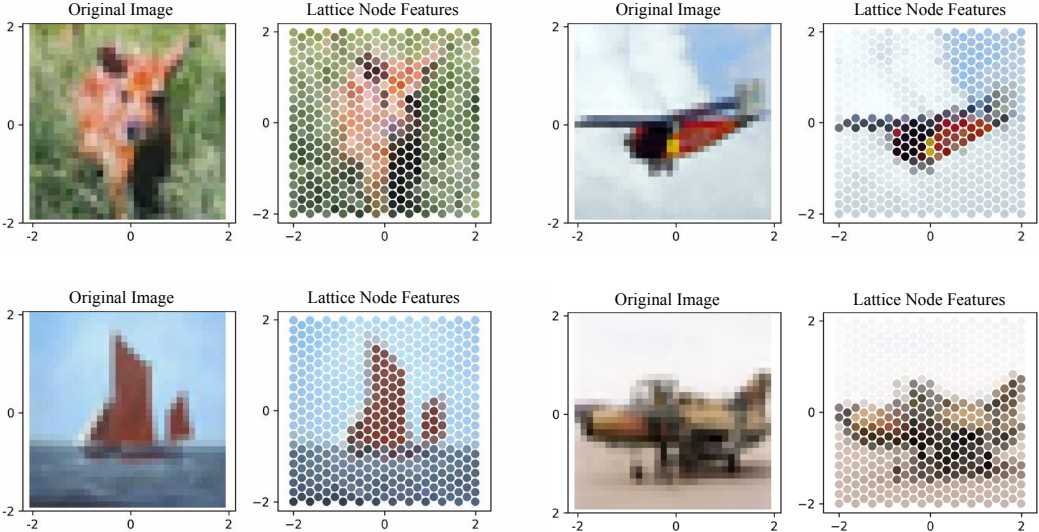

Figure H.2: Examples of triangular lattice graphs with node features derived from CIFAR image pixel values. The CIFAR image is shown on the left, and the corresponding lattice-mapped node features is depicted on the right. In practice, the feature of each node is a 3 dimensional vector of RGB values. As described in the main text, output node features are generated by propagating these input features through the directed triangular lattice for 10 hops. The node regression task involves predicting these output node features from the input node features.

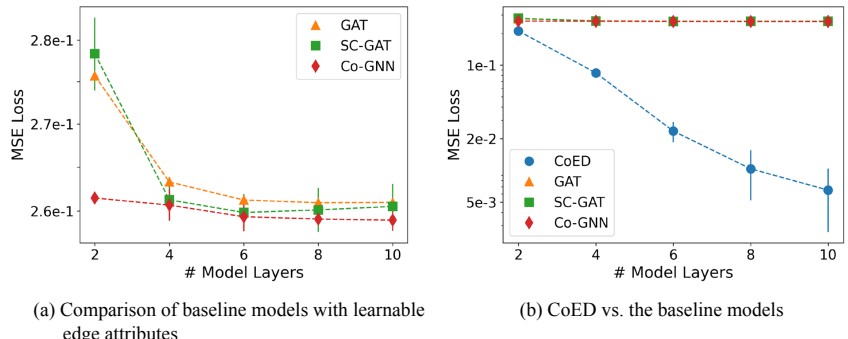

(a) Comparison of baseline models with learnable edge attributes

(b) CoED vs. the baseline models

Figure H.3: Left: Mean Squared Error (MSE) loss as a function of model depth for GAT, the self-consistent GAT formulation (SC-GAT), and Cooperative GNN (Co-GNN) applied to the node regression task on a triangular lattice graph with node features derived from CIFAR images. Right: Same as left, but including results from CoED GNN. CoED GNN is the only model that demonstrates consistent performance improvement with increasing depth.

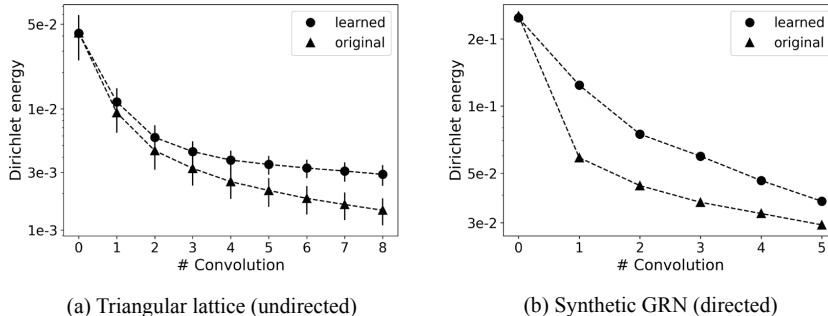

(a) Triangular lattice (undirected)       (b) Synthetic GRN (directed)

Figure H.4: Dirichlet energy of node features computed after applying the graph Laplacian over multiple iterations (number of convolutions shown on the x-axis) for both the original graph Laplacian and the Laplacian constructed using the learned edge directions from CoED. Results for the triangular lattice graph with CIFAR-derived node features are shown on the left, and results for synthetic GRN data are shown on the right. The slower decay of Dirichlet energy when using the Laplacian with learned edge directions indicates that CoED mitigates the oversmoothing problem.

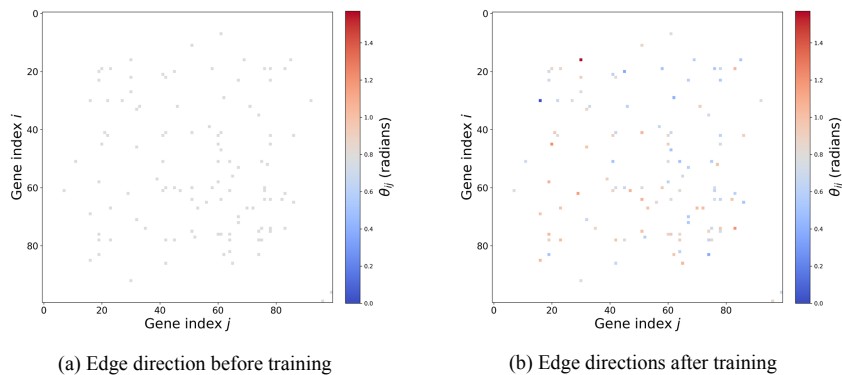

(a) Edge direction before training       (b) Edge directions after training

Figure I.1: Visualization of the Perturb-seq data graph before and after applying CoED to learn edge directions. Initially, all edges are undirected. CoED assigns continuous edge directions, revealing a complex and structured pattern. For clarity, a subset of genes is visualized, with the $i$-$j$ element of the matrix representing the phase angle assigned to each edge using our fuzzy Laplacian formulation.

H.3 compares the performance of CoED GNN and a baseline GAT model as a function of model depth. CoED GNN's performance improves with increasing depth, whereas GAT's performance saturates, showing that CoED GNN's learned edge directions alleviate the oversmoothing problem. This trend was also observed in Figure 4 with the random features. To further illustrate this, we computed the Dirichlet energy of node features by applying the graph Laplacian multiple times to simulate information flow over increasing numbers of hops. As shown in Figure H.4, when the original graph Laplacian is used, the Dirichlet energy decreases rapidly, indicating that node features are converging to the same value. In contrast, using the Laplacian of the directed graph with learned edge directions results in a slower decrease in Dirichlet energy, showing that oversmoothing is mitigated by directed information flow.

# I VISUALIZATION OF THE LEARNED EDGE DIRECTIONS

To illustrate the edge directions learned by CoED, we visualized the phase angles of edges in a real-world single-cell Perturb-seq dataset (see main text for details). As shown in Figure I.1, the initial edge directions are undirected, with all phases set to $\pi/4$. After training, CoED uncovers a complex structure of continuous edge directions, capturing the gene-gene interactions that connect the nodes (genes) in the graph.

