# OpenReview forum: "Improving Graph Neural Networks by Learning Continuous Edge Directions"
_ICLR.cc/2025/Conference — ICLR 2025 Poster_

### Official Review · Reviewer_xqhX · 2024-10-19

**Soundness:** 3
**Presentation:** 3
**Contribution:** 3
**Rating:** 8
**Confidence:** 4

**Summary:**

This paper proposes a novel framework for undirected and directed graphs called Continuous Edge Direction (CoED) GNN, based on the insight to assign fuzzy edge directions in a novel complex-valued Laplacian matrix. The paper includes empirical comparison to various GNNs on both synthetic and real-world datasets, and theoretically shows the expressive power of CoED.

**Strengths:**

- The paper is generally well-written and well-motivated.
- The topics of expressive power, directed GNNs, and oversmoothing in GNNs are significant.
- The proposal of fuzzy edge directions is novel and effective.
- The statements are theoretically and empirically grounded.

**Weaknesses:**

- No source code is provided.
- It would be better to show the runtime comparison explicitly, e.g. via a table, when comparing CoDE with MagNet etc.
- The paper uses the idea of separate aggregation of in- and out- information in addition to self. [1] does something similar and is based on flow imbalance in its objective. It would be better to add some discussion and comparison in experiments.
- In line 315, it is not clear where $\alpha$ should appear in the formula.
- Table 1 compares a CoED version without edge direction learning, how about with edge direction learning for node classification?
- (bonus) It would be better to show the ultimate directions (visualized) for some directed or undirected graphs to show more insights.

Reference:

[1] He, Y., Reinert, G., & Cucuringu, M. (2022, December). DIGRAC: digraph clustering based on flow imbalance. In Learning on Graphs Conference (pp. 21-1). PMLR.

**Questions:**

- What is the runtime comparison explicitly, e.g. via a table, when comparing CoDE with MagNet etc.?
- The paper uses the idea of separate aggregation of in- and out- information in addition to self. [1] does something similar and is based on flow imbalance in its objective. How are they compared empirically?
- In line 315, where should $\alpha$ appear in the formula?
- Table 1 compares a CoED version without edge direction learning, how about with edge direction learning for node classification?

Reference:

[1] He, Y., Reinert, G., & Cucuringu, M. (2022, December). DIGRAC: digraph clustering based on flow imbalance. In Learning on Graphs Conference (pp. 21-1). PMLR.

---

> ### Author Response · Authors · 2024-11-24
> **Response**
>
> We thank the reviewer for the invaluable feedback and suggestions. We are delighted that the reviewer found our paper well-written and well-motivated, and our topics significant. We are also glad that the reviewer found our proposal of fuzzy edge directions novel and effective and our statements grounded both theoretically and empirically.
>
> We address in detail the weaknesses and the questions raised by the reviewer.
>
> > No source code is provided.
>
> We thank the reviewer for raising this point. We now include the source code for our analysis: https://anonymous.4open.science/r/coed-gnn-0347
>
>
> > What is the runtime comparison explicitly, e.g. via a table, when comparing CoED with MagNet etc.?
>
> The reviewer raises an important point. In the revised manuscript in Supplementary Section I, we provide extensive runtime comparison between CoED and the baseline models. Since the only additional source of computes in CoED are the in- and out- fuzzy propagation matrices (Eq. 2 in the manuscript), CoED shows comparable runtime to DirGNN. Notably, CoED achieves much faster forward and backward pass time compared to Magnet or GAT even when we perform layerwise edge directions learning—where we compute propagation matrices from different phase angles at each layer.
>
> > The paper uses the idea of separate aggregation of in- and out- information in addition to self. [1] does something similar and is based on flow imbalance in its objective. It would be better to add some discussion and comparison in experiments.
>
> We thank the reviewer for bringing this relevant reference to our attention. This paper uses the concept of flow imbalance to assign nodes of directed graphs to clusters. While we cannot do a direct empirical comparison to this node clustering method, as our learning tasks are either node classification or regression, we believe our continuous edge direction learning can provide a more finely-tuned imbalance profile for this algorithm to pick up on. Clustering directed graphs is an important area of research, and we are interested in exploring the application of our continuous edge direction learning in this direction. In the revised manuscript, we now include a discussion of reference [1] and how it conceptually relates to our work.
>
> > In line 315, where should α appear in the formula?
>
> We have clarified this in the revised manuscript.
>
> > Table 1 compares a CoED version without edge direction learning, how about with edge direction learning for node classification?
>
> The reviewer raises an important point. In Table 1, we benchmark performance of node classification against baseline models without learning the edge directions. First, this is done because our main goal for Table 1 is to directly compare our proposed fuzzy Laplacian, and the corresponding architecture, to alternative forms of Laplacian, such as the magnetic Laplacian. Second, because CoED learns continuous edge directions by directly optimizing for the learning task and without relying on node features, it requires access to all of the nodes of the graph during the training. Masking a subset of nodes as test nodes would mean that the directions of the edges connected to the test nodes will not be learned during training. In fact, the directions of those edges will be tuned to improve the performance of the training nodes at the expense of the test nodes, resulting in diminished performance. Therefore, CoED is best suited for graph ensemble data (for example, the synthetic and real-world data sets used in our paper) where the multiple realization of node features and their corresponding target values exist for the same graph and thus we do not need to mask the part of a graph during the training. In the revised manuscript, we have added additional text to make this point abundantly clear.
>
> > (bonus) It would be better to show the ultimate directions (visualized) for some directed or undirected graphs to show more insights.
>
> We thank the reviewer for this helpful suggestion. We have included a visualization of the learned continuous edge directions for the Perturb-seq data (an originally undirected graph, where nodes are genes, and edges are gene-gene interactions) in the revised manuscript in the newly added Section H of the supplement.
>
> Again, we thank the reviewer for the tremendously helpful comments and feedback. We hope that we have addressed all remaining concerns of the reviewer. We are of course happy to answer any further questions.

---

> > ### Comment · Reviewer_xqhX · 2024-11-25
> >
> > I hereby acknowledge reading the rebuttal.

---

> > > ### Author Response · Authors · 2024-11-26
> > >
> > > We again thank the reviewer for evaluating our work and the helpful suggestions.

---

### Official Review · Reviewer_SbM9 · 2024-11-01

**Soundness:** 3
**Presentation:** 2
**Contribution:** 2
**Rating:** 5
**Confidence:** 4

**Summary:**

This paper leverages "fuzzy edges" to improve the flow of information across a graph. CoED GNN learns directions (coefficients) for each edge. This is achieved through a complex-valued fuzzy Laplacian. Experiments on both synthetic and real-world datasets demonstrate CoED GNN's good performance in node classification and regression tasks, attributed to its ability to optimize long-range feature propagation.

**Strengths:**

1. The fuzzy Laplacian enables CoED GNN to handle directional message passing, preserving the discriminative features over longer distances.
2. CoED GNN shows performance gains in diverse datasets, highlighting its adaptability to both synthetic and real-world applications.

**Weaknesses:**

1. The edge directions are essentially edge weights. GAT can also learn edge weights, and it even leverages node features while doing so, unlike CoED GNN. Since GAT utilizes more information, it should perform better. At the very least, the weights learned by GAT should not be worse than those learned by CoED GNN. It is puzzling that GAT performs worse than CoED GNN in the experimental results provided by the authors. The authors should provide theoretical provements to show that CoED GNN is superior.
2. The authors should explain why CoED GNN can alleviate the problem of homogenization of information across the graph. Also, the authors should  provide experimental validations.

**Questions:**

1. Could the authors provide theoretical analysis of the superiority of CoED GNN over GAT?
2. In Equation (1), why $(L_F)_{ij}=\text{exp}(i\theta_{ij})$? Is the first $i$ the imaginary unit? If so, the authors should use different letter to denote the $i$th node.
3. Could the authors provide explanation and experimental results to show that CoED GNN can alleviate the problem of homogenization of information across the graph?

---

> ### Author Response · Authors · 2024-11-24
> **Response part 1**
>
> We thank the reviewer for the helpful feedback and suggestions. We are glad that the reviewer found the proposed fuzzy Laplacian for long-range directional message-passing to be a strength of our paper and highlighted the observed performance gains across diverse data sets.
>
> We now address the questions and weaknesses raised by the reviewer.
>
> > Could the authors provide theoretical analysis of the superiority of CoED GNN over GAT?
>
> The reviewer raises an important point. It is indeed true that GAT can also learn edge weights. However, GAT does so solely based on the features of the nodes spanned by the edge. Our proposed method learns the continuous edge directions to optimize the learning task directly and can learn edge directions even when the node features are not informative. This is why, CoED outperforms GAT in our synthetic data sets and the real data sets. Importantly, to learn edge directions independently from the node features, we need data sets of graph ensembles where multiple realizations of the nodes (and their corresponding regression target values) are provided on the same graph during training. To make this point abundantly clear, in the revised manuscript, we have included a new Supplementary Section (Appendix G). There, we propose a toy model of a linear graph that forms a closed loop. The input node features are random (and therefore uninformative). The output node features are the same as the input features but shifted by one node across the graph in the clockwise direction. GAT cannot learn the optimal direction of information transmission because the features contain no information about this direction. CoED on the other hand directly learns the edge directions to optimize the regression task independent of the node features and can therefore correctly predict the output values. We hope this example clarifies why CoED GNN is superior to GAT for certain classes of problems. Our evaluation of real-world graph ensemble datasets demonstrate that this is true for many important real-world problems.
>
>
> > In Equation (1), why $(L_F){ij}=\text{exp}(i\theta{ij})$ ?Is the first i the imaginary unit? If so, the authors should use different letter to denote the $i$th node.
>
> The reviewer is absolutely correct. The first i in the exponent is meant to denote an imaginary unit. We have changed them to the upright style character $\mathrm{i}$ in the revised manuscript.

---

> ### Author Response · Authors · 2024-11-24
> **Response part 2**
>
> > Could the authors provide explanation and experimental results to show that CoED GNN can alleviate the problem of homogenization of information across the graph?
>
> We thank the reviewer for this important question. In the revised manuscript, in Supplementary Section G, we provide an explanation and empirical evidence why CoED GNN alleviates the homogenization (or oversmoothing problem) on graphs. When a graph is undirected, aggregating information across the neighbors of a node repeatedly is akin to diffusion of information across the nodes. This is because when node i sends information to node j, node j also sends information to node i. Diffusion prohibits long-range information transmission across the graph and results in a rapid convergence of the node features to the averaged feature across all the nodes. In contrast, directed edges allow information to flow instead of diffuse. This is because node i can now send information to node j without receiving information in return (if there is a directed edge from node i to node j). This flow of information allows long-range information transmission without excessive averaging across nodes alleviating both the oversmoothing problem and improving the performance in tasks that require long-range information transmission. To demonstrate this experimentally, in Figure G.3 of the revised manuscript, we plot the performance of CoED as a function of model depth, which corresponds to the number of hops used to transmit information, for the synthetic dataset of the triangular lattice with node features obtained from CIFAR images. Compared to the baseline of GAT, CoED shows continued improvement in performance with increasing model depth, indicating that oversmoothing has been alleviated. To show this even more directly, we also compute the Dirichlet energy of the node features on this graph and plot it as a function of number of times the graph Laplacian is applied to the node features (again equivalent to the number of times information is transmitted by hopping across neighboring nodes). When the undirected graph Laplacian is used, the node features rapidly converge to a fixed value and exhibit low Dirichlet energies. In contrast, when the directed graph Laplacian is used, computed from the learned edge directions, the Dirichlet energy reduces less rapidly, indicating that homogenization is occurring more slowly.
>
> We thank the reviewer again for extremely helpful feedback and suggestions. We are of course happy to answer any additional questions. If there are no further questions, we kindly ask the reviewer to consider re-evaluating our manuscript under a new light and potentially upgrading the score if appropriate.

---

> ### Author Response · Authors · 2024-12-02
> **Follow-up on Revised Manuscript for Discussion Period**
>
> Dear Reviewer SbM9,
>
> As the ICLR discussion period draws to a close today, we wanted to thank you once again for your thoughtful feedback, which has been invaluable in refining our work. We have carefully considered and addressed all your comments in the revised version of our paper.
>
> We believe that the updated manuscript (especially the newly added Supplemental Section G) effectively addresses the points you raised and significantly improves the clarity and rigor of our work.
>
> We kindly ask you to review the revised version if time permits. Your insights and further engagement would be greatly appreciated as the discussion period concludes.
>
> Thank you again for your time and expertise.
>
> Best regards,
> The authors

---

### Official Review · Reviewer_QZRJ · 2024-11-03

**Soundness:** 3
**Presentation:** 3
**Contribution:** 2
**Rating:** 3
**Confidence:** 5

**Summary:**

The paper presents a novel graph neural network architecture named Continuous Edge Direction (CoED) GNN, which improves upon traditional GNNs by introducing the concept of fuzzy edge directions. These directions allow for more effective feature propagation by varying continuously, enabling more controlled information flow between nodes. CoED GNN utilizes a complex-valued Laplacian that represents these fuzzy directions, where the real and imaginary parts correspond to opposite directions of information flow. The architecture is especially effective in graph ensemble settings, where the structure is static but node features vary, such as in gene regulatory networks and power grids.

**Strengths:**

1. The paper is well-written, and the motivation of the model is reasonable.
2. The idea of learning optimal edge directions, and using phase angles to scale the direction-aware information flow is intuitive and interesting.
3. Theoretical Analysis appears to be rigorous and formally describes the expressive power of the proposed model

**Weaknesses:**

1. Directed graph Laplacians have been extensively explored in previous studies, including seminal works like those by F. Chung [1] and on Hermitian Laplacian [2]. The introduction of the fuzzy graph Laplacian is a significant contribution of this work. However, while comparisons have been made with the magnetic Laplacian, it would be beneficial to extend these comparisons to include contemporary directed graph Laplacians to fully contextualize its advancements and differences.

2. From my perspective, employing phase angles to dictate continuous edge directions is designed to adaptively modulate the weights from in-neighbors and out-neighbors. However, could we alternatively employ a shared learnable function $f$, , such as, $\mathrm{Sigmoid}(f(u_i, u_j))$ and $1- \mathrm{Sigmoid}(f(u_i, u_j))$ , to scale the information flow? This approach might offer a simpler alternative to using complex values. Furthermore, related work such as superGAT [3] enhances GAT by identifying important neighbors, a concept that could potentially be extended to directed graphs. By incorporating the separate message-passing strategy of DirGNN—which independently considers in-edges and out-edges—we could apply superGAT to each direction separately. Subsequently, the two representations could be combined to produce the final node representation.

3. While the concept of learning optimal edge directions is intriguing, I find the experimental results presented in the paper unconvincing. Firstly, as illustrated in Table 1, the CoED model achieves the best results in only one out of the five datasets examined. Secondly, the paper omits experiments on several pertinent datasets commonly used in related works, notably those employed by DirGNN, which is frequently referenced in this study. Specifically, there are no results included for datasets such as SNAP-PATENTS, ROMAN-EMPIRE, and ARXIV-YEAR, despite DirGNN reporting significant findings on these along with various other baselines. Incorporating these datasets could enhance the persuasiveness of the results. Thirdly, I view SAGE as a specific instantiation of DirGNN, implying that the performance of DirGNN should be at least as good, if not better, than that of SAGE. To enhance DirGNN's performance, particularly on the Texas and Wisconsin datasets, I experimented with incorporating an MLP layer following the final layer of DirGNN. This modification has significantly improved its performance. Additionally, the Texas and Wisconsin datasets are relatively small, and the Chameleon and Squirrel datasets reportedly contain significant amounts of duplicate data [4]. Therefore, I believe these datasets may not be the most suitable for evaluating the model's performance. The authors might consider using the AM-Photo, AM-Computer, or Citeseer datasets, as employed in [5], to provide a more robust assessment.



[1] Laplacians and the Cheeger inequality for directed graphs.

[2] Graph signal processing for directed graphs based on the hermitian laplacian.

[3] How to find your friendly neighborhood: Graph attention design with self-supervision.

[4] A critical look at the evaluation of GNNs under heterophily: Are we really making progress?

[5] Digraph Inception Convolutional Networks.

**Questions:**

See Weaknesses

---

> ### Author Response · Authors · 2024-11-24
> **Response part 1**
>
> We thank the reviewer for extremely helpful feedback and suggestions. We are delighted that the reviewer found the paper well-written, the model reasonable, and the idea of using continuous edge directions for enhancing information flow on graphs intuitive and interesting. We are also glad that the reviewer found our theoretical analysis used to formally describe the expressivity of our model to be rigorous.
>
> We now address the weaknesses raised by the reviewer in detail.
>
> > Directed graph Laplacians have been extensively explored in previous studies, [...] it would be beneficial to extend these comparisons to include contemporary directed graph Laplacians to fully contextualize its advancements and differences.
>
> We agree with the important point raised by the reviewer. In the revised manuscript, we have extended our comparisons to other types of Laplacians for directed graphs besides the magnetic Laplacian. The Hermitian Laplacian [2] is indeed the magnetic Laplacian. However, the Laplacian introduced by Chung [1], where a Hermitian Laplacian is constructed from the transition matrix of the graph, is an alternative Laplacian for directed graphs. To our knowledge, the Chung Laplacian and the magnetic Laplacian are the only other Laplacians proposed for directed graphs. We now compare the performance of the Chung Laplacian when applied to node classification (revised Table 1) to that of the fuzzy Laplacian. In addition, we have also included a comparison of the Chung Laplacian to the fuzzy Laplacian when applied to our synthetic data sets (revised Table 2). In all cases, we observe an improved performance when the fuzzy Laplacian is used.
>
> > From my perspective, employing phase angles to dictate continuous edge directions is designed to adaptively modulate the weights from in-neighbors and out-neighbors. However, could we alternatively employ a shared learnable function  f, , such as, Sigmoid(f(ui,uj)) and 1−Sigmoid(f(ui,uj)), to scale the information flow? [...] Furthermore, related work such as superGAT [3] enhances GAT by identifying important neighbors, a concept that could potentially be extended to directed graphs. [...]
>
> The reviewer raises an interesting point. It is possible to learn edge weights based on the features of the two nodes connected by an edge, for example in the form of Sigmoid(f(ui,uj)), and then impose that outgoing and incoming messages are consistent by requiring that the edge weight in the opposite direction is given by 1−Sigmoid(f(ui,uj)). However, such edge weights, although consistent with the edge directions, are still functions of the features of the nodes. The key distinction of our method is that the continuous edge directions are learned directly through optimization of the learning task over the entire graph and not as a function of node features. Our synthetic data set of the triangular lattice graph with random node features (Figure 3 and Table 2), where node features are completely uninformative, illustrates this most clearly. To demonstrate this empirically, we implemented the reviewer's suggestion and constructed an attention-like model as proposed by the reviewer: edge weights were computed from the node features with the constraint that the learned edge weights in opposite directions sum to one. We then applied this model to the new synthetic data of the triangular lattice graph with node features obtained from CIFAR images. In this dataset, the features of the two nodes connected by an edge are mostly identical and therefore it would be difficult to learn informative edge-weights from these features. For example, the pair of node features, {$(u_i, u_j) | j \in \mathcal{N}(i) $}, and the resulting edge weights {$f(ui, uj) | j \in \mathcal{N}(i) $} may be all identical. Therefore, the model with the explicit correspondence between i→j and i←j directions, which we call “self-consistent (SC)-GAT”, could still suffer if features are uninformative. We added the discussion of CoED vs. GAT in the supplemental section G of the revised manuscript, and empirically demonstrate that CoED outperforms SC-GAT in Figure G.3. CoED outperforms this model because it can learn edge directions without relying on the node features.

---

> ### Author Response · Authors · 2024-11-24
> **Response part 2**
>
> > Furthermore, related work such as superGAT [3] enhances GAT by identifying important neighbors, a concept that could potentially be extended to directed graphs. [...]
>
>
> The idea of extending the superGAT [3] framework to directed graphs is interesting and insightful. superGAT extends GAT by predicting whether an edge should exist between two nodes based on the features of those nodes. As described above, this approach is still conceptually different from our method and more limited because it relies on the edge features to modulate the attention between two neighboring nodes. Although extending superGAT to directed graphs is beyond the scope of our manuscript, to test this idea empirically, we applied a similar approach called [1] Cooperative GNN (or Co-GNN) to our synthetic data set where node features are not informative for learning the edge directions. Similar to superGAT that predicts whether an edge should exist between two nodes based on their features, Co-GNN uses the features of a node and that of its neighbors to classify a node as one that either broadcasts to its neighbors (which amounts to making outgoing edges), listens to its neighbors (incoming edges), does both, or neither. As shown in Figure G.3, CoED outperforms Co-GNN because it can learn the optimal edge directions without relying on node features.
>
>
> > While the concept of learning optimal edge directions is intriguing, I find the experimental results presented in the paper unconvincing. [...]. The authors might consider using the AM-Photo, AM-Computer, or Citeseer datasets, as employed in [5], to provide a more robust assessment.
>
> We thank the reviewer for excellent suggestions on applying CoED to larger and more representative data sets. In the revised manuscript, we have done exactly as the reviewer suggested. 1) We have evaluated CoED’s performance on Roman-Empire, SNAP-patents, and Arxiv-Year. As shown in revised Table 1, CoED outperforms the other baseline models in these datasets . 2) As suggested by the reviewer, we have incorporated the SAGE instantiation of DirGNN so that DirGNN can either be run with GCN-, GAT-, or SAGE- convolution depending on which performs better. In the revised Table 1, DirSageConv was chosen from hyperparameter search for Texas, Wisconsin, Citeseer, AM-Photo, and AM-Computers. 3) In the revised manuscript, we now use the filtered versions of Chameleon and Squirrel data sets to address the reviewer’s concern. CoED’s ranking compared with the other baselines did not change. 4) Finally, we now also include benchmarking on the AM-Photo, AM-Computer, and Citeseer datasets as suggested by the reviewer. CoED is again one of the top models on these datasets. We hope these additional experiments address the reviewer’s concern with benchmarking for the node classification task. We would also like to emphasize that the main result of our paper is not that CoED outperforms existing models in node classification tasks. Rather, we include Table 1 to provide empirical evidence that our proposed architecture using the fuzzy Laplacian can outperform Magnet (which uses the Hermitian Laplacian) and the newly added architecture based on the Chung Laplacian. We have revised the text to reflect this point more clearly.
>
> We hope that we have addressed all the concerns raised by the reviewer. We are happy to answer any additional questions. If there are no further questions, we kindly ask the reviewer to consider re-evaluating our paper in a new light and potentially consider upgrading their score.
>
> [1] Finkelshtein, Ben, et al. "Cooperative graph neural networks." ICML 2024.

---

> ### Author Response · Authors · 2024-12-02
> **Follow-up on Revised Manuscript for Discussion Period**
>
> Dear Reviewer QZRJ,
>
> As the discussion period for ICLR concludes today, we wanted to reach out and thank you again for your valuable feedback, which has been instrumental in refining our work. We have carefully addressed all your comments and incorporated the suggested changes into the revised version of our paper.
>
> We believe the updated manuscript thoroughly resolves the concerns you raised and enhances the clarity and impact of our work (see in particular, the revised Table 1, and the newly added Supplemental Section G).
>
> We would greatly appreciate it if you could take a moment to review the revisions. Your expertise and further input would be invaluable as the review process comes to a close.
>
> Thank you again for your thoughtful engagement.
>
> Best regards,
> The authors

---

### Official Review · Reviewer_b7qy · 2024-11-05

**Soundness:** 3
**Presentation:** 2
**Contribution:** 3
**Rating:** 6
**Confidence:** 3

**Summary:**

Traditional GNNs often suffer from feature oversmoothing due to their diffusion-like message-passing mechanism over undirected graphs. To address this, the authors propose the Continuous Edge Direction (CoED) GNN framework that assigns differentiable fuzzy edge directions to graph edges, allowing features to preferentially flow in one direction and introducing a complex-valued Laplacian for directed graphs with fuzzy edges.

**Strengths:**

The paper is well-written and offers an intriguing evaluation on graph ensemble data, where the graph structure remains fixed but multiple realizations of node features and targets exist—an important task that demands attention. Additionally, the authors demonstrate that their method is more expressive than the Magnet Laplacian.

**Weaknesses:**

1. The paper omits important baselines and related work, such as [3], which share similar concepts and should be included for a comprehensive comparison.
2. If the theorem is highlighted in the abstract and introduction, it should be mentioned explicitly in the main paper for consistency and emphasis.
3. The evaluation on node classification tasks is unconvincing:
   - Only one homophilous dataset is included in the study.
   - The Chameleon and Squirrel datasets used are known to be problematic [1], and it is advisable to use filtered versions for more reliable results.
   - The heterophilous datasets used for evaluation are all quite small, and the results do not approach the current state-of-the-art. To more robustly validate the claims, experiments on larger-scale heterophilous datasets [1][2] are necessary.


References

[1] Platonov, O., Kuznedelev, D., Diskin, M., Babenko, A., & Prokhorenkova, L. (2023). A critical look at the evaluation of GNNs under heterophily: Are we really making progress?. arXiv preprint arXiv:2302.11640.

[2] Derek Lim, Felix Hohne, Xiuyu Li, Sijia Linda Huang, Vaishnavi Gupta, Omkar Bhalerao, and Ser Nam Lim. Large scale learning on non-homophilous graphs: New benchmarks and strong simple methods. Advances in Neural Information Processing Systems, 34, 2021.

[3]Finkelshtein, Ben, et al. "Cooperative graph neural networks." ICML 2024.

**Questions:**

1. What is the runtime of the proposed methods, and how does it compare to the baselines? Providing a detailed runtime analysis would enhance the evaluation.
2. For the node classification task, how does the performance change if the edge direction is made learnable? Prior work, such as [3], suggests that it will likely improve performance.

---

> ### Author Response · Authors · 2024-11-24
> **Response part 1**
>
> We thank the reviewer for tremendously helpful feedback and suggestions. We are pleased that the reviewer found the paper well-written, the evaluation on graph ensemble data intriguing, and highlighted our expressivity comparison to magnetic Laplacian.
>
> Below we address the weaknesses and questions raised by the reviewer.
>
> > The paper omits important baselines and related work, such as [3], which share similar concepts and should be included for a comprehensive comparison.
>
> We thank the reviewer for this suggestion. In the revised manuscript, we have included multiple additional baselines, including reference [3] (Cooperative GNNs, or Co-GNNs) suggested by the reviewer. We include Co-GNN as a baseline both in the node classification task, where edge directions are not learned (revised Table 1), and also apply it to our synthetic triangular lattice data (newly added Supplemental Section G). In both node classification and our synthetic data, CoED outperforms Co-GNNs. This is because although Co-GNN assigns distinct roles to each node (such as listen, broadcast, do both, or neither), it does so relying on the features of a node and its neighbors. CoED, in contrast, learns the continuous edge directions by directly optimizing for the learning task and can therefore learn optimal edge directions even when the node features are uninformative. This is particularly apparent in the new synthetic triangular lattice problem where node features are created from the RGB pixel values of images in the CIFAR dataset. Here, most neighbors have identical features which are completely uninformative for learning edge directions. Lastly, the continuous edge directions allow for more complex information transmission across graphs than assigning one of four possible roles to each node.
>
>
> > If the theorem is highlighted in the abstract and introduction, it should be mentioned explicitly in the main paper for consistency and emphasis.
>
> We thank the reviewer for this suggestion. In the revised manuscript, we explicitly mention the theorem in the main text.
>
>
> > The evaluation on node classification tasks is unconvincing:
> Only one homophilous dataset is included in the study.
> The Chameleon and Squirrel datasets used are known to be problematic [1], and it is advisable to use filtered versions for more reliable results.
> The heterophilous datasets used for evaluation are all quite small, and the results do not approach the current state-of-the-art. To more robustly validate the claims, experiments on larger-scale heterophilous datasets [1][2] are necessary.
>
> We agree with the reviewer on all of these points and have addressed them in the revised manuscript. 1) We now have included Citeseer, AM-Photo, AM-Computers as additional homophilic data sets in Table 1. 2) We have replaced the Chameleon and Squirrel datasets with the filtered versions as suggested by the reviewer in the revised table. The conclusion that CoED performs well on these data sets relative to the baselines still holds. 3) In the revision, we have included multiple larger-scale heterophilic datasets. These are Roman-Empire, Arxiv-Year, and SNAP-Patents. Importantly, CoED consistently ranks as one of the best performing models on these data sets, highlighting the benefits of our proposed architecture and the fuzzy Laplacian for node classification on heterophilic graphs.

---

> ### Author Response · Authors · 2024-11-24
> **Response part 2**
>
> > What is the runtime of the proposed methods, and how does it compare to the baselines? Providing a detailed runtime analysis would enhance the evaluation.
>
> In the revised manuscript, we provide a table of runtimes for CoED and representative baseline models in Supplementary Section I. The only additional computes in CoED are due to the in- and out- fuzzy propagation matrices (Eq. 2 in our paper) during the forward pass. Except for layerwise edge direction learning, CoED needs to compute them only once so it adds minimal overhead compared to DirGCN which shares the similar AGGREGATE and UPDATE steps. Even when learning edge directions at each layer separately, CoED achieves faster forward and backward pass times compared to other baseline models like GAT or Magnet.
>
>
> > For the node classification task, how does the performance change if the edge direction is made learnable? Prior work, such as [3], suggests that it will likely improve performance.
>
> We thank the reviewer for raising an important point. Because CoED does not learn the edge directions from the features of the nodes (as is done by GAT or Co-GNN), it needs access to all of the nodes of the graph during training. If a subset of the nodes are set aside for testing then the directions of the edges incident to the test nodes will not be learned and in fact will be tuned to optimize classification of the training nodes at the expense of the test nodes. Empirically, we observe a decrease in performance when a subset of nodes are masked as test nodes and edge directions are learned (as in the node classification task in Table 1). Our method is most useful when applied to Graph ensemble data where the graph structure is held constant across multiple realizations of node features and corresponding target values. In this setting, we split the realizations of features and targets. Therefore we can use the entire nodes to learn the edge directions without the need for masking a subset of nodes as test nodes. As shown in Table 2 and Table 3 of the manuscript, learning edge directions in graph ensemble data significantly improves performance. We have added text to the revised manuscript to make this point more clear.
>
> Again, we thank the reviewer for their invaluable feedback and insightful comments. We hope that our answers above address the reviewer’s concerns. We are of course more than happy to answer any additional questions. If there are no further questions, we kindly ask the reviewer to consider reevaluating our work from a fresh perspective and potentially upgrading their score.

---

> > ### Comment · Reviewer_b7qy · 2024-11-28
> >
> > Thank you for your response. I've read all your reply and revised manuscript. I would like to increase my rating.

---

> > > ### Author Response · Authors · 2024-11-29
> > >
> > > We again thank the reviewer for their insightful feedback and for reevaluating our work.

---

### Author Response · Authors · 2024-11-24
**Global Response**

We sincerely thank all the reviewers for their invaluable feedback and insightful suggestions. We are delighted that the reviewers found our paper “well-written” (**b7qy**, **QZRJ**, **xqhX**) and “well-motivated” (**QZRJ**, **xqhX**). The reviewers noted that our paper “offers an intriguing evaluation on graph ensemble data…an important task that demands attention” (**b7qy**) and that the “the topics of expressive power, directed GNNs, and oversmoothing in GNNs are significant” (**xqhX**). The reviewers also wrote that the idea of learning continuous edge directions is “intuitive and interesting” (**QZRJ**) and that the “proposal of fuzzy edge directions is novel and effective” (**xqhX**). The reviewers also highlighted our theoretical analysis, stating that it is “rigorous and formally describes the expressive power of the proposed model” (**QZRJ**) and that the “statements are theoretically and empirically grounded” (**xqhX**). Finally, the reviewers highlighted our empirical results as well, writing, “CoED GNN shows performance gains in diverse datasets, highlighting its adaptability to both synthetic and real-world applications” (**SbM9**).

We have significantly improved our manuscript based on the reviewers’ feedback and suggestions and have uploaded a revised version. The revisions are marked in blue. We address all the individual comments below, but the key improvements to the paper are summarized here:

- We have significantly increased the benchmarking of the node classification task in Table 1. We have included additional datasets, including Roman-Empire, Arxiv-Year, SNAP-patents, Citeseer, AM-Photo, AM-Computers, and filtered versions of Squirrel and Chameleon (**b7qy**, **QZRJ**) and have included additional baselines such as Cooperative GNN (**b7qy**), SAGE implementation of DirGNN, and additional forms of Laplacians for directed graphs (**QZRJ**).
- In Supplemental Section G, we describe how CoED GNN is different from GAT (**SbM9**), provide empirical evidence that CoED GNN outperforms a modified GAT architecture where edge weights are tuned to be consistent across incoming and outgoing messages (**QZRJ**), and provide empirical evidence that CoED GNN alleviates the problem of feature homogenization (**SbM9**).
- We have included a comprehensive run time analysis in Supplemental Section I (**b7qy**, **xqhX**).
- In Supplemental Section H, we provide a visualization of the learned continuous edge directions (**xqhX**).


We thank the reviewers again for their thoughtful consideration of our manuscript. We hope that the revised manuscript and our replies to the individual reviewers below have addressed all the concerns and questions. We look forward to answering any additional questions.

---

### Comment · Area_Chair_tLzQ · 2024-12-01
**Reminder: Please Review Author Responses**

Dear Reviewers,

As the discussion period is coming to a close, please take a moment to review the authors’ responses if you haven’t done so already. Even if you decide not to update your evaluation, kindly confirm that you have reviewed the responses and that they do not change your assessment.

Thank you for your time and effort!

Best regards, AC

---

### Meta-Review · Area_Chair_tLzQ · 2024-12-07

**Metareview:**

The paper presents a novel framework for graph neural networks (GNNs) that introduces fuzzy edge directions using a complex-valued Laplacian. This approach addresses key challenges in traditional GNNs, such as feature oversmoothing and limited expressiveness for directed graphs. The proposed CoED GNN enhances information flow between nodes by assigning differentiable edge directions, enabling controlled and adaptive message passing. This framework shows promise, particularly in scenarios involving static graph structures but varying node features, such as graph ensemble tasks. The authors provide a solid theoretical foundation for the expressiveness of their model and demonstrate its effectiveness on both synthetic and real-world datasets.

**Strengths**

Novelty: The introduction of fuzzy edge directions and a complex-valued Laplacian is innovative and provides a fresh perspective on improving GNN performance.

Empirical Evaluation: The evaluation includes both synthetic and real-world datasets, showcasing the model's adaptability across diverse scenarios.

Relevance to Graph Ensemble Tasks: The paper addresses graph ensemble tasks, a valuable yet underexplored application area in graph learning.

**Weaknesses**
While the reviewers raised concerns regarding the selection of datasets (especially commonly used heterophilic ones) and baseline comparisons, I find these points less critical given the primary application focus on graph ensemble tasks. However, I identified a few areas for improvement that could strengthen the final version:

Potential Overfitting and Task Limitations. Assigning a freely trainable parameter to each edge introduces the risk of overfitting, particularly in tasks outside the graph ensemble domain where the number of parameters might far exceed the available labels. This concern aligns with the authors' response to Reviewer b7qy, where masking nodes resulted in notable performance degradation. I suggest to emphasize the primary focus on graph ensemble tasks in the introduction/motivation and explicitly discuss the model's limitations on commonly used heterophilic graphs. This clarification would help future researchers better understand the scope of the approach.

Expressive Power and WL Test Comparisons. The proposed framework's expressive power comparisons with standard GNNs and magnetic Laplacians, framed through the lens of WL tests, require clarification. The framework's edge-specific parameters inherently break permutation symmetry, which the WL tests assume to preserve. I suggest to address this distinction explicitly in the final version and provide a more extensive comparison with magnetic Laplacians, particularly regarding expressive power. Additionally, several relevant works on magnetic Laplacians were not discussed. I recommend incorporating those fundamental references to provide a more comprehensive discussion:

1. MA Shubin. Discrete Magnetic Laplacian. Communications in Mathematical Physics, 1994.

2. Zhang et al. Magnet: A Neural Network for Directed Graphs. NeurIPS 2021.

3. Fanuel et al. Magnetic Eigenmaps for Community Detection in Directed Networks. Physical Review E, 2017.

4. Fanuel et al. Magnetic Eigenmaps for the Visualization of Directed Networks. Applied and Computational Harmonic Analysis, 2018.

5. Furutani et al. Graph Signal Processing for Directed Graphs Based on the Hermitian Laplacian. ECML PKDD 2019.

6. Geisler et al. Transformers Meet Directed Graphs. ICML 2023.

7. Huang et al. What Are Good Positional Encodings for Directed Graphs? arXiv 2024.


Despite the identified weaknesses, I believe the paper's strengths in novelty and relevance outweigh its limitations. The innovative contributions and targeted applications suggest a promising direction for future research, leading me to lean toward acceptance.

**Additional Comments On Reviewer Discussion:**

During the rebuttal stage, the authors effectively addressed the concerns raised by the two negative reviewers by clearly distinguishing their work from other edge-weight parameterized models, such as GAT. Additionally, they provided further empirical evaluations to strengthen their case. As noted earlier, the proposed model is naturally well-suited for graph ensemble tasks but less effective on commonly used heterophilic graphs. This limitation is understandable, but I strongly recommend that the authors explicitly clarify this point in the final version to avoid any potential confusion for future researchers.

---

### Decision · Program_Chairs · 2025-01-22

Accept (Poster)